# Human-in-the-Loop Policy Optimization for Preference-Based Multi-Objective Reinforcement Learning

**Tianmeng Hu** [1 2]   **Biao Luo** [3]   **Ke Li** [1 2]

## Abstract

Multi-objective reinforcement learning (MORL) seeks policies that effectively balance conflicting objectives. However, presenting many diverse policies without accounting for the decision maker's (DM's) preferences can overwhelm the decision-making process. On the other hand, accurately specifying preferences in advance is often unrealistic. To address these challenges, we introduce a human-in-the-loop MORL framework that interactively discovers preferred policies during optimization. Our approach proactively learns the DM's implicit preferences in real time, requiring no a priori knowledge. Importantly, we integrate this preference learning directly into a parallel optimization framework, balancing exploration and exploitation to identify high-quality policies aligned with the DM's preferences. Evaluations on a complex quadrupedal robot simulation environment demonstrate that, with only $40$ interactions, our proposed method can identify policies aligned with human preferences, e.g., running like a dog. Further experiments on seven MU-JOCO tasks and a multi-microgrid system design task against eight state-of-the-art MORL algorithms fully demonstrate the effectiveness of our proposed framework. Demonstrations and full experiments are in our supplemental website.

## 1. Introduction

Many real-world decision-making tasks involve multiple objectives, which often conflict. Examples include balancing makespan and energy consumption in workflow schedul-ing (Qin et al., 2020), optimizing microgrid designs for both grid stability and user benefits (Xu et al., 2021), and navigating trade-offs among fuel cost, efficiency, and safety in robotic control (Xu et al., 2020). Multi-objective reinforcement learning (MORL) addresses these problems by learning policies that simultaneously consider such conflicting criteria (Liu et al., 2015; Hayes et al., 2022). Existing MORL approaches can be broadly classified into two main categories.

The first, and perhaps most prevalent, category aggregates multiple objective functions into a single scalar reward, usually through linear scalarization (Gábor et al., 1998; Mannor & Shimkin, 2001). Each objective receives a weight based on the decision maker's (DM's) prior preferences, and standard reinforcement learning (RL) algorithms then optimize this aggregated scalar reward (Duan et al., 2016; Chen et al., 2021). While conceptually straightforward, this approach relies heavily on the DM's ability to accurately specify their preferences beforehand. In real-world tasks, however, preferences are often implicit and difficult to articulate numerically. Even when clear preference information is available, designing an effective weighted reward is challenging because the relationship between reward weights and policy outcomes is often nonlinear and unpredictable (Van Moffaert et al., 2014; Hayes et al., 2022). As a result, DMs often rely on iterative trial-and-error to adjust the reward weights until an acceptable policy emerges.

The second major MORL strategy aims to identify a diverse set of Pareto-optimal policies spanning various trade-offs (Natarajan & Tadepalli, 2005; Ikenaga & Arai, 2018; Chen et al., 2019; Yang et al., 2019; Hu et al., 2023; Basaklar et al., 2023). This set is then presented to the DM, who selects the most suitable policies. Although this method avoids requiring initial preferences, it often generates hundreds of policies when only a few are practically relevant. Producing and evaluating these irrelevant policies consumes considerable computational resources. For example, when training a bipedal robot, algorithms might discover Pareto-optimal but impractical gaits. While these solutions satisfy Pareto optimality in the objective space, they offer little practical value. Thus, pursuing diversity without guidance can waste substantial computational effort, potentially hindering

[1]Living Systems Institute, University of Exeter, United Kingdom [2]Department of Computer Science, University of Exeter, United Kingdom [3]School of Automation, Central South University, China. Correspondence to: Biao Luo <biao.luo@hotmail.com>, Ke Li <k.li@exeter.ac.uk>.

*Proceedings of the $43^{rd}$ International Conference on Machine Learning*, Seoul, South Korea. PMLR 306, 2026. Copyright 2026 by the author(s).

the discovery of genuinely useful policies.

Unlike the aforementioned paradigms, some approaches, such as interactive Q-steering (Vamplew et al., 2017), incorporate human guidance within an interactive MORL (iMORL) framework. In interactive Q-steering, the algorithm first presents an estimated Pareto front (PF) to the DM, who then specifies preferred regions or targets. The algorithm uses this feedback to adjust its policy search direction dynamically. This approach allows the DM to iteratively refine preferences during training, removing the need for precise reward weights defined upfront. While interactive Q-steering, for example, might require an initial PF estimation, the broader concept of dynamically guiding the MORL search based on evolving, elicited preferences is powerful. However, iMORL frameworks that actively learn and adapt to evolving DM preferences to guide the exploration of the policy space have received limited attention (Roijers et al., 2017; 2018; Peschl et al., 2022), despite their potential to reduce unnecessary exploration and computational effort.

To address these challenges, we propose PBMORL, a general iMORL framework. PBMORL balances the targeted nature of single-policy methods with the exploratory capabilities of multi-policy approaches. It interactively learns the DM's preferences, removing the need for precise a priori preference articulation or extensive reward shaping. A key innovation lies in how learned preferences are utilized. While policy diversity is beneficial for escaping local optima and finding better solutions (Hu & Luo, 2024), PBMORL does not prematurely constrain the search based on these preferences. Instead, it integrates preference learning into the multi-policy optimization process by dynamically guiding resource allocation. Preferences bias the selection and refinement of candidate policies. This approach steers the search toward regions of the PF aligned with evolving DM interests, inherently balancing the exploitation of known preferences with the exploration of new regions. PBMORL refines its policies through human-in-the-loop feedback, progressively identifying solutions of genuine practical relevance. Our framework consists of three interconnected modules: ❶ `Seeding Promising Policies`: generates an initial, diverse set of policies approximating the PF. ❷ `Preference Elicitation`: actively queries the DM to obtain qualitative feedback and translates it into a quantitative model of the DM's utility. ❸ `Policy Optimization`: uses the utility model from elicitation to guide a multi-policy RL algorithm. This guidance directs the parallel optimization of candidate policies, ensuring alignment with DM preferences.

We first evaluated PBMORL on a challenging UNITREE GO2 quadrupedal robot task (Unitree, 2024), optimizing for speed and energy efficiency. Compared to extensively tuned standard RL approaches, PBMORL produces significantly

more natural and stable robot behaviors aligned directly with the DM's implicit preferences, i.e., behaviors difficult to encode manually. In particular, under high-speed preference, PBMORL matches the top baseline's speed but reduces average torque by $37.4\%$. Under a low-torque preference, PBMORL achieves a $79.0\%$ torque reduction. Additional extensive evaluations on MUJOCO benchmarks (Todorov et al., 2012) and a multi-microgrid design problem (Chiu et al., 2015) confirm PBMORL's consistent superiority over eight state-of-the-art MORL methods.

## 2. Preliminaries

In this paper, a MORL problem is formulated as a 5-tuple $\langle \mathcal{S}, \mathcal{A}, T, \mathbf{r}, \boldsymbol{\gamma} \rangle$ multi-objective Markov decision process (MOMDP):

- $\mathcal{S}$ is the state space, i.e., the set of all available $n$-dimensional states $\mathbf{s} \in \mathbb{R}^n$.

- $\mathcal{A}$ is the action space, i.e., the set of all available $l$-dimensional actions $\mathbf{a} \in \mathbb{R}^l$.

- $T(s' \mid s, a)$ is the transition probability function.

- $\mathbf{r} = (r_1(s_t, a_t), \dots, r_m(s_t, a_t))^\top$ is the reward vector.

- $\boldsymbol{\gamma} = (\gamma_1, \dots, \gamma_m)^\top \in (0, 1]^m$ is a vector of discount factors.

In a MOMDP, a policy $\pi : \mathcal{S} \rightarrow \mathcal{A}$ determines how the current state $s_t$ move to the next one $s_{t+1}$ by taking the action $a_t \sim \pi(s_t)$. In particular, $\pi$ is associated with a vector of expected returns $\mathbf{J}(\pi) = (J_1(\pi), \cdots, J_m(\pi))^\top$ where $J_i(\pi) = \mathbb{E}\left(\sum_{t=0}^{H} \gamma_i^t r_i(s_t, a_t)\right)$. Accordingly, a multi-objective policy optimization (MOPO) problem is defined as:

$$\text{maximize} \quad \mathbf{F}(\pi) = (f_1(\pi), \cdots, f_m(\pi))^\top \quad (1)$$

where $\pi$ is a policy and $\mathbf{F}(\pi) = \mathbf{J}(\pi)$ is an objective vector. Objectives are assumed to be conflicting with each other. That said, improving one objective often degrades others.

**Definition 2.1.** Given two policies $\pi^1$ and $\pi^2$, $\pi^1$ is said to *dominate* $\pi^2$ (denoted by $\pi^1 \succeq \pi^2$) if and only if $f_i(\pi^1) \geq f_i(\pi^2)$ for all $i \in \{1, \cdots, m\}$ and $\mathbf{F}(\pi^1) \neq \mathbf{F}(\pi^2)$.

**Definition 2.2.** A policy $\pi^*$ is said to be *Pareto-optimal* if and only if $\nexists \pi'$ such that $\pi' \succeq \pi^*$.

**Definition 2.3.** Let $\mathcal{P}$ denote the feasible policy space. The set of all Pareto-optimal policies is called the *Pareto-optimal set* (PS), i.e., $\mathcal{PS} = \{\pi^* \in \mathcal{P} \mid \nexists \pi' \in \mathcal{P} : \pi' \succeq \pi^*\}$, and their corresponding objective vectors form the *Pareto-optimal front* (PF), i.e., $\mathcal{PF} = \{\mathbf{F}(\pi^*) \mid \pi^* \in \mathcal{PS}\}$.

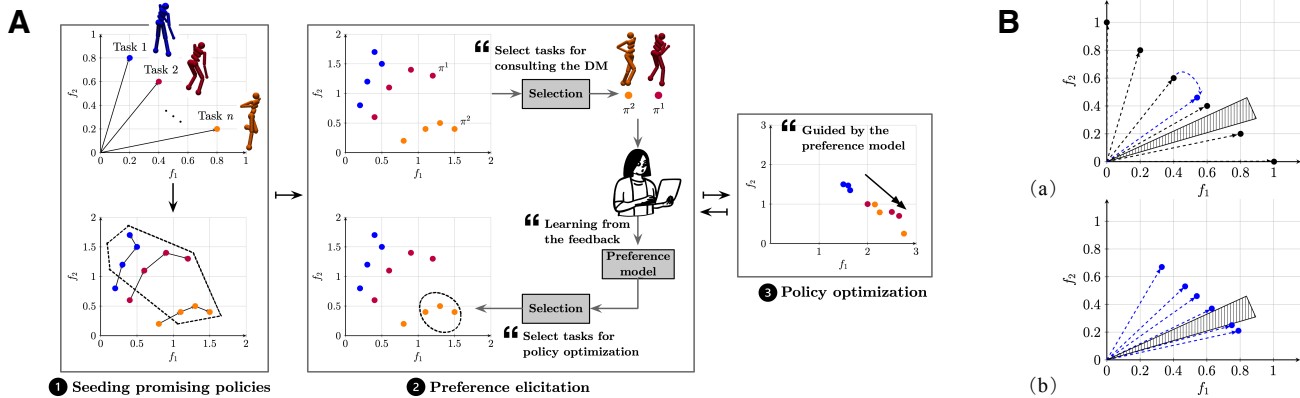

*Figure 1.* **A**. Flowchart of PBMORL. It iterates between the preference elicitation and the policy optimization modules until the stopping criterion is met, and outputs the preferred policies. **B**. Illustration of the weight vector adjustment in Step 4. ($a$) The region of interest (ROI) is highlighted as the shaded cone region versus the evenly distributed weight vectors (denoted as ●). ($b$) Adjusted weight vectors towards the ROI (denoted as ●).

## 3. Proposed Method

The central idea of PBMORL is to learn a utility model from a small number of pairwise comparisons, and then use this model to allocate optimization effort within a multi-policy search. We do not assume the DM's preference can be represented by a "true" scalarization weight vector. Human preferences are often nonlinear and behavior-level, and therefore cannot be captured by a fixed linear weighting of objectives. Moreover, even if a DM could specify a weight vector, optimizing the corresponding scalarized reward does not reliably produce the intended behavior in RL due to non-convexity and sensitivity to exploration and initialization.

PBMORL therefore treats preferences as an interactive specification signal. It learns a nonlinear utility function over objective vectors $J(\pi)$, but does not directly use this utility as the per-policy RL objective. Instead, scalarization serves only as a search parameterization: the learned utility is translated into an adaptive distribution over scalarized subproblems, thereby reallocating optimization effort toward preference-consistent regions. As shown in Fig. 1, PBMORL alternates between preference elicitation and policy optimization: the elicitation module actively queries policy pairs, fits a utility model with calibrated uncertainty, and translates the learned preference into a task distribution that biases subsequent optimization toward a preference-consistent region of interest while retaining exploration. The optimization module then improves a set of candidate policies in parallel under these tasks, producing higher-quality options for the next elicitation round. Pseudocode is provided in Appendix A.2.

### 3.1. Seeding promising policies

PBMORL starts from randomly initialized policies, such policies are uninformative for preference elicitation since com-

parisons between two poor behaviors provide little signal about the DM's intent. We therefore begin with a short Seeding stage that bootstraps a diverse set of reasonably competent policies before entering the human-in-the-loop iterations.

Following the decomposition principle of MOEA/D (Zhang & Li, 2007), we decompose the MOMDP into $N$ scalarized subproblems. Each subproblem combines objectives linearly using a weight vector:

$$\tilde{J}(\pi, \mathbf{w}) = \sum_{i=1}^{m} w_i J_i(\pi), \qquad (2)$$

where $\mathbf{w} = (w_1, \ldots, w_m)^\top$ defines the trade-off among the $m$ objectives, satisfying $\sum_{i=1}^{m} w_i = 1$. We construct an evenly distributed set of weights $\mathcal{W} = \{\mathbf{w}^i\}_{i=1}^{N}$ on the unit simplex using the Das–Dennis method (Das & Dennis, 1998), yielding a balanced coverage of objective trade-offs. Each weight specifies one scalarized RL objective. In our implementation, we refer to a policy together with its associated optimization weight as a *task*, denoted by $\tau = (\pi, \mathbf{w})$. These tasks are optimized independently in parallel.

We allocate the first $5\%$ of the total training budget to seeding (see sensitivity analysis in Appendix B.8). At the end of the seeding phase, we obtain a diverse set of non-dominated policies, denoted by $\Pi$, which will serve as the initial candidate pool for the subsequent preference elicitation rounds.

### 3.2. Preference elicitation

The Preference Elicitation module is the interface between the DM and the multi-policy optimizer. Its role is to (i) collect a small amount of qualitative feedback with minimal DM burden, (ii) fit a calibrated utility model that predicts the DM's preference over candidate policies, and (iii) convert this model into actionable guidance that

reallocates optimization effort toward preference-consistent regions while preserving exploration. Throughout, we maintain a current candidate set $\Pi_c$ produced by the optimization module, and a preference dataset $\mathcal{D}$ accumulated across rounds.

### 3.2.1. CONSULTATION

The `Consultation` step collects lightweight feedback from the DM via pairwise comparisons. Given a policy pair $\langle \pi^1, \pi^2 \rangle$, the DM indicates whether $\pi^1$ is *better*, *worse*, or *indifferent* to $\pi^2$, denoted by $\pi^1 \succ \pi^2$, $\pi^1 \prec \pi^2$, or $\pi^1 \simeq \pi^2$, respectively. Pairwise feedback is typically easier and more consistent than assigning absolute scores across diverse behaviors.

Since querying all pairs is infeasible, we actively select informative policies from the current pool $\Pi_c$. Inspired by active learning and multi-armed bandits (Auer, 2002), we define an acquisition score:

$$\mathtt{I}(\pi) = \mu(\pi) + \alpha \sqrt{\sigma(\pi)\tilde{n}/\tilde{n}_\pi}, \qquad (3)$$

where $\mu(\pi)$ and $\sigma(\pi)$ are the posterior mean and variance predicted by the preference model (see Section 3.2.2). Here, $\tilde{n}$ is the total number of queries made so far and $\tilde{n}_\pi$ is the number of times $\pi$ has been queried. The parameter $\alpha > 0$ trades off selecting policies that are promising (high $\mu$) versus uncertain or under-explored (high $\sigma$ or small $\tilde{n}_\pi$).

In each consultation round, we query $\pi^1 = \arg\max_{\pi \in \Pi_c} I(\pi)$ and $\pi^2 = \arg\max_{\pi \in \Pi_c \setminus \{\pi^1\}} I(\pi)$. The DM's response on $(\pi^1, \pi^2)$ is then added to the preference dataset $\mathcal{D}$ for training the preference model.

### 3.2.2. PREFERENCE LEARNING

Given the preference dataset $\mathcal{D} = \{(\pi_i^1, \pi_i^2, o_i)\}_{i=1}^{|\mathcal{D}|}$ with $o_i \in \{\succ, \prec, \simeq\}$, we learn a latent utility function $u(\mathbf{J}(\pi)) : \mathbb{R}^m \to \mathbb{R}$ that is consistent with the DM's comparisons. Let $\tilde{\Pi} = \{\tilde{\pi}^j\}_{j=1}^\kappa$ denote the set of unique policies involved in $\mathcal{D}$, and define the latent utility vector $\mathbf{u}_{\tilde{\Pi}} = \left( u(\mathbf{J}(\tilde{\pi}^1)), \ldots, u(\mathbf{J}(\tilde{\pi}^\kappa)) \right)^\top$. Following standard practice (Chu & Ghahramani, 2005), we model noisy comparisons using a probit likelihood. Let $\Phi(\cdot)$ denote the standard normal cumulative distribution function. For a comparison label $o_i = \succ$, the likelihood is

$$\begin{aligned} &\mathbb{P}\big(o_i = \succ \mid u(\mathbf{J}(\pi_i^1)), u(\mathbf{J}(\pi_i^2))\big) \\ &= \Phi\left( \frac{u(\mathbf{J}(\pi_i^1)) - u(\mathbf{J}(\pi_i^2))}{\sqrt{2}} \right), \end{aligned} \qquad (4)$$

and symmetrically for $o_i = \prec$ by swapping the two policies. Ties ($\simeq$) are handled by assuming nearly equal utilities within a small margin.

We place a zero-mean Gaussian Process (GP) prior with an RBF kernel on $u(\cdot)$, which induces $\mathbf{u}_{\tilde{\Pi}} \sim \mathcal{N}(\mathbf{0}, K_{\tilde{\Pi}})$,

where $K_{\tilde{\Pi}}$ is the kernel matrix on $\tilde{\Pi}$. We estimate utilities via the maximum a posteriori (MAP) solution $\mathbf{u}^\star = \arg\max_{\mathbf{u}_{\tilde{\Pi}}} \mathbb{P}(\mathbf{u}_{\tilde{\Pi}})\mathbb{P}(\mathcal{D} \mid \mathbf{u}_{\tilde{\Pi}})$, equivalently minimizing the negative log-posterior:

$$\begin{aligned} \mathbf{u}^\star = \arg\min_{\mathbf{u}_{\tilde{\Pi}}} \; &-\sum_{i=1}^{|\mathcal{D}|} \log \mathbb{P}\big(o_i \mid u(\mathbf{J}(\pi_i^1)), u(\mathbf{J}(\pi_i^2))\big) \\ &+ \frac{1}{2}\mathbf{u}_{\tilde{\Pi}}^\top K_{\tilde{\Pi}}^{-1} \mathbf{u}_{\tilde{\Pi}}. \end{aligned} \qquad (5)$$

This objective can be solved efficiently using iterative methods such as Newton–Raphson (Chu & Ghahramani, 2005).

After obtaining $\mathbf{U} = \mathbf{u}^\star$, we predict the utility of any policy $\pi$ under the GP posterior. Let $\mathbf{k}_\pi = \left[ k(\pi, \tilde{\pi}^1), \ldots, k(\pi, \tilde{\pi}^\kappa) \right]^\top$. The posterior mean $\mu(\pi)$ and predictive variance $\sigma(\pi)$ are given by

$$\mu(\pi) = \mathbf{k}_\pi^\top K_{\tilde{\Pi}}^{-1} \mathbf{U}, \qquad (6)$$

$$\sigma(\pi) = k(\pi, \pi) - \mathbf{k}_\pi^\top \left( K_{\tilde{\Pi}} + \Lambda^{-1} \right)^{-1} \mathbf{k}_\pi, \qquad (7)$$

where $\Lambda$ captures uncertainty around the MAP estimate. These $\mu(\pi)$ and $\sigma(\pi)$ are used in the `Consultation` module for active querying and in `Preference translation` for constructing $\psi(\pi) = \mu(\pi) + \beta\,\sigma(\pi)$.

### 3.2.3. PREFERENCE TRANSLATION

Once an estimate of the DM's utility is obtained, we use it to construct a weighted multi-objective reward and apply RL for policy improvement. To mitigate convergence to local optima, we express preference guidance as a distribution over tasks rather than a single fixed weight vector. The translation step uses the learned utility model to identify an ROI on the weight simplex and then constructs a mixture of exploitation and exploration tasks that focuses computation inside this ROI while maintaining coverage.

We first score each candidate policy $\pi \in \Pi_c$ by an uncertainty-aware preference score $\psi(\pi) = \mu(\pi) + \beta\,\sigma(\pi)$, where $\beta \geq 0$ controls optimism under uncertainty. We select the top $\kappa_1$ policies under $\psi(\cdot)$, denoted by $\dot{\Pi}$, and collect their associated weight vectors $\dot{W}$ (the weights under which these policies were optimized). Intuitively, $\dot{W}$ provides a data-driven localization of the weight-space region that is currently consistent with the DM's feedback.

To allocate more optimization effort near the ROI without collapsing diversity, we generate a denser reference set of weights $\tilde{W}$ on the simplex (using the same construction as in Sec. 3.1), and then bias each $\tilde{\mathbf{w}} \in \tilde{W}$ toward the ROI by moving it toward its nearest ROI anchor:

$$\mathbf{w}^r = \arg\min_{\mathbf{w} \in \dot{W}} \|\mathbf{w} - \tilde{\mathbf{w}}\|_2, \quad \bar{\mathbf{w}} = (1 - \eta)\tilde{\mathbf{w}} + \eta\,\mathbf{w}^r, \quad (8)$$

where $\eta \in (0, 1]$ controls how aggressively we concentrate around the ROI. We denote the resulting set of biased

weights by $\bar{W} = \{\bar{\mathbf{w}} | \tilde{\mathbf{w}} \in \tilde{W}\}$. This operation reshapes the task distribution: it increases sampling density near $\dot{W}$ while still retaining a spread of directions.

Finally, we define the task set for the next optimization round as the union of $\kappa_1$ exploitation tasks and $\kappa_2$ exploration tasks. Exploitation tasks refine promising solutions by pairing each $\dot{\pi} \in \dot{\Pi}$ with its original weight in $\dot{W}$. Exploration tasks launch new searches using randomly selected biased weights $\bar{\mathbf{w}} \in \bar{W}$, initialized from a nearby promising policy in $\dot{\Pi}$. This mixture allocates more effort to preference-consistent regions while still probing nearby trade-offs to correct misspecification and avoid premature convergence.

The resulting task set is passed to the `Policy Optimization` module to produce an updated candidate pool $\Pi_c$ for the next elicitation round. A systematic analysis of the hyperparameters $\kappa_1$, $\kappa_2$, $\beta$, and $\eta$ is provided in Appendix B.4 and Appendix B.7.

### 3.3. Policy optimization

The `Policy Optimization` module executes one round of multi-policy improvement under the task set produced by `Preference translation`. Each task is a tuple $\tau = (\pi_{\text{seed}}, \mathbf{w})$, where $\mathbf{w}$ specifies a scalarized MORL objective and $\pi_{\text{seed}}$ provides a warm start. For any task, we optimize the scalarized return $\tilde{J}(\pi, \mathbf{w}) = \mathbf{w}^\top \mathbf{J}(\pi)$. In our implementation, we adopt a PPO-based optimizer and run all tasks in parallel for a fixed environment-interaction budget per round.

After optimizing all tasks, we obtain a set of updated policies $\Pi_{\text{new}}$ and add them to the candidate pool. We then update $\Pi_c$ by retaining the non-dominated policies (and their associated training weights) for the next elicitation round: $\Pi_c \leftarrow \text{ND}(\Pi_c \cup \Pi_{\text{new}})$. The updated $\Pi_c$ is passed back to the `Preference Elicitation` module to generate more informative comparisons and refine the preference-consistent region in subsequent rounds.

## 4. Experiments

We evaluate PBMORL in 9 different environments, including a UNITREE GO2 quadrupedal robot control task (Unitree, 2024), seven MUJOCO (Todorov et al., 2012) robotic control benchmarks, and a multi-microgrid system design (MMSD) problem (Chiu et al., 2015).

### 4.1. Experiments on UNITREE GO2 robot control

This is a challenging high-dimensional continuous control task with a 48-dimensional state space and a 12-dimensional action space (target joint angles). Such complexity makes it difficult to manually design effective reward functions. Thus, it specifically tests PBMORL's capability to directly learn effective policies aligned with user preferences in a demanding, high-dimensional environment.

#### 4.1.1. EXPERIMENTAL SETTINGS

To formulate a multi-objective evaluation, we define two conflicting objectives: maximizing forward velocity $r_v$ and minimizing energy consumption $r_e$. The corresponding reward functions are:

$$\begin{cases} r_v = v_f + r_{\text{alive}}, \\ r_e = -C_\tau \sum_i \tau_i^2 + r_{\text{alive}} + C, \end{cases} \quad (9)$$

where $v_f$ is forward velocity, $r_{\text{alive}}$ is a survival reward, $\tau_i$ is the $i$-th torque, $C_\tau$ scales the torque penalty, and $C$ ensures positive rewards.

Our primary baseline is scalarized PPO. We tested this baseline with two reward shaping schemes. The first, `basic reward` version, directly uses the objectives from equation (9). It calculates a weighted sum:

$$r_{\text{basic}} = C_v r_v + C_e r_e, \quad (10)$$

where $C_v$ and $C_e$ are weight coefficients applied to $r_v$ and $r_e$, respectively. The second scheme, `complex reward` adds auxiliary shaping terms $r_{\text{add}}$ to $r_e$. This is formulated as:

$$r_{\text{complex}} = C_v r_v + C_e (r_e + r_{\text{add}}), \quad (11)$$

$r_{\text{add}}$ (detailed in Appendix D.1) guide the robot's posture and penalize instabilities like tilting or collisions. Such shaping is often necessary for standard RL in complex tasks. For both scalarized PPO schemes, we evaluated five different weight combinations, as shown in Table 1.

Note that, for PBMORL, we do not employ any reward shaping. Instead, we introduce real human feedback and consider three distinct preference profiles: ▶ **Stationary:** Prioritize minimal movement (zero velocity). ▶ **Moderate speed:** Balance forward motion with energy efficiency. We set the target speed for medium-speed preference to $2.5\ m/s$. ▶ **High speed:** Maximize forward velocity while maintaining stability. Beyond quantifiable metrics, humans often exhibit implicit preferences that cannot be expressed via weights, such as a "natural" gait.

#### 4.1.2. EXPERIMENTAL RESULTS

We trained PBMORL for each preference setting and scalarized PPO for each weight combination. Each PBMORL run and each scalarized PPO run used the same environment-interaction budget of $10^7$ steps. We evaluated policies by averaging cumulative rewards over five independent test runs, calculated using equation (9) for consistency. Fig. 2A visualizes the policies' performance in objective space. Although PBMORL often finds several similar policies per preference,

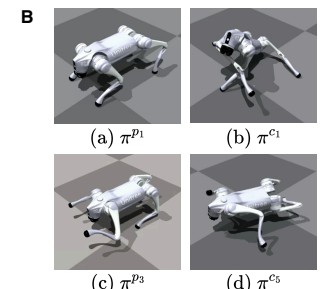
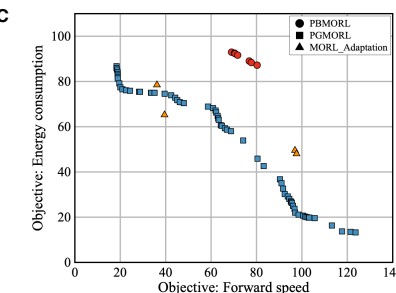

*Figure 2.* **A**. Visualization of policies obtained by PBMORL and scalarized PPO under different preference and reward weight settings. **B**. Snapshots of different policies during evaluation. **C**. Comparison with conventional multi-policy MORL algorithms under the moderate speed preference.

*Table 1.* Average forward speed and torque of the policies trained via PBMORL and scalarized PPO during testing. These policies correspond to those shown in Fig. 2.

| | Preference | Average Speed (m/s) ↑ | Average Torque (N · m) ↓ |
|---|---|---|---|
| PBMORL | $\pi^{p_1}$ : Stationary | 0.0039 | 6.2749 |
| | $\pi^{p_2}$ : Moderate speed | 2.6005 | 13.6821 |
| | $\pi^{p_3}$ : High speed | 5.5442 | 28.4631 |
| Scalarized PPO basic reward | $\pi^{b_1}$ : $[0.1, 0.9]$ | 0.0652 | 29.9237 |
| | $\pi^{b_2}$ : $[0.3, 0.7]$ | 0.4785 | 31.2369 |
| | $\pi^{b_3}$ : $[0.5, 0.5]$ | 0.1372 | 31.2468 |
| | $\pi^{b_4}$ : $[0.7, 0.3]$ | 4.7869 | 47.3405 |
| | $\pi^{b_5}$ : $[0.9, 0.1]$ | 4.3275 | 47.0493 |
| Scalarized PPO complex reward | $\pi^{c_1}$ : $[0.1, 0.9]$ | 0.0059 | 43.4544 |
| | $\pi^{c_2}$ : $[0.3, 0.7]$ | 0.0023 | 40.5503 |
| | $\pi^{c_3}$ : $[0.5, 0.5]$ | 3.9753 | 46.3568 |
| | $\pi^{c_4}$ : $[0.7, 0.3]$ | 4.9380 | 46.8985 |
| | $\pi^{c_5}$ : $[0.9, 0.1]$ | 5.6801 | 45.4319 |

we present only one representative policy for clarity, as performance differences within these groups were minimal. Table 1 lists average speed and torque for these representative policies. Key findings include:

1. **PBMORL aligns effectively with preferences.** Policies $\pi^{p_1}$ to $\pi^{p_3}$ accurately matched intended behaviors. Specifically, $\pi^{p_1}$ kept the robot stationary, $\pi^{p_2}$ achieved a moderate speed efficiently ($2.60\,\mathrm{m/s}$), and $\pi^{p_3}$ reached a higher speed ($5.54\,\mathrm{m/s}$) with increased energy usage. Visually, PBMORL's policies ($\pi^{p_1}$, $\pi^{p_3}$) exhibited more natural and stable postures than scalarized PPO alternatives (see Figure 2B and some demonstration videos from the supplemental website).

2. **PBMORL consistently achieves high performance.** In the objective space (Figure 2A), PBMORL's solutions consistently dominated almost all scalarized PPO policies, except for $\pi^{c_5}$.

3. **Scalarized PPO struggles to balance objectives.** Despite experimenting with two reward formulations and

five weight combinations, scalarized PPO often converged prematurely to suboptimal regions in objective space. This result suggests scalarized PPO easily becomes trapped in local optima. PBMORL's guided exploration enabled it to better avoid these pitfalls.

4. **Preference guidance facilitates the discovery of superior non-dominated policies.** As shown in Figure 2C, conventional MORL methods such as PGMORL, which explore the Pareto front without user preference guidance, tend to yield a diverse but suboptimal policy set. This wastes effort on regions irrelevant to the user's preference and fails to identify the true "sweet spot". Note that we allocate a $5\times$ larger training budget to baselines designed to recover the full Pareto front, which further demonstrates the efficiency of PBMORL.

5. **Reward weights are unreliable proxies for final behaviors.** The scalarized PPO results show that increasing the nominal weight on one objective does not necessarily produce a monotonic improvement in the corresponding measured behavior. This practical non-monotonicity

is expected in deep RL because policy optimization is non-convex and sensitive to exploration and initialization. PBMORL reduces the need for manual weight tuning by using pairwise preference feedback to guide the search.

### 4.2. Experiments on MuJoCo and MMSD

We next compared PBMORL against four state-of-the-art preference-based MORL algorithms: MORL-Adaptation(Yang et al., 2019), META-MORL(Chen et al., 2019), MOMPO(Abdolmaleki et al., 2020), and MORAL(Peschl et al., 2022). This comparison included seven diverse MuJoCo control tasks (Todorov et al., 2012) and an MMSD problem (Chiu et al., 2015). Further details on benchmarks and algorithms are in Appendix D. For these large-scale benchmarks, we use simulated preference oracles rather than real-time human annotations.

Evaluating preference-based MORL requires specialized metrics, as standard measures may not capture alignment with DM intent (Li et al., 2018b). We adopt the *approximation accuracy* metric, defined as the distance to an ideal "golden policy": $\epsilon^\star(\Pi) = \min_{\pi \in \Pi} \|\mathbf{J}(\pi) - \mathbf{J}(\pi^\star)\|_2$, where $\Pi$ is the algorithm's output set of non-dominated policies. $\pi^\star$ is the ideal policy according to the DM's preferences (unknown to algorithms). Details on defining $\pi^\star$ are provided in Appendix C.4.

From the selected results in Table 2 and Figure 3, we have two key observations: ▶ On complex tasks (Ant-v2, Hopper-v3, Humanoid-v2, MMSD), PBMORL consistently outperformed other methods. Often, PBMORL found policies that dominated all those identified by peer algorithms. ▶ On simpler tasks, PBMORL performed comparably to the best alternative methods. We hypothesize these tasks have simpler PFs. Therefore, standard methods without deliberately being navigated to ROI can still find solutions located in the targeted regions. Full results for all eight tasks are in Appendix C.2, Table 11, and Figure 10.

### 4.3. Ablations and additional analyses

**Impact of interaction frequency**  PBMORL elicits DM preferences through pairwise policy comparisons. Our main experiments used 40 such interactions. To examine sensitivity to this parameter, Appendix B.1 explores fewer interactions. Results show that PBMORL maintains strong performance even with significantly fewer interactions, achieving good outcomes with as few as 10 to 20 queries.

**Ablation study of the GP model**  We selected a GP for the Preference Learning module because it offers two key advantages. First, GPs are highly data-efficient, ideal for learning from limited human feedback, where we only have 40 pairwise comparison results. Second, GPs naturally pro-

vide uncertainty estimates. This uncertainty helps PBMORL strategically select informative policy pairs for DM consultation, effectively balancing exploration and exploitation. Here we conducted an ablation study by replacing the GP with two alternative preference learning methods. Results in Appendix B.2 showed that the GP-based approach performed best. Nevertheless, because PBMORL is a general framework to involve human in the loop of MORL, we believe GP is not a mandatory choice while new preference learning model can be an interesting future direction.

**PBMORL scales to many-objective scenarios**  Most MORL algorithms for complex continuous-control robotics tasks focus on only 2-3 objectives, largely due to scalability challenges. We go beyond this regime by examining the scalability of PBMORL to higher-dimensional objective spaces. In Appendix C.1, we design and conduct a 5-objective experiment on the challenging Unitree Go2 task, which demonstrates that PBMORL maintains strong performance even in many-objective settings.

**Comparison with conventional multi-policy MORL**  Conventional multi-policy MORL algorithms typically aim to approximate the entire PF. In contrast, PBMORL specifically targets optimal policies aligned with the DM's implicit preferences, which only cover a smaller region of the PF. In this experiment, we compared PBMORL with leading multi-policy methods like PGMORL (Xu et al., 2020) and PDMORL (Basaklar et al., 2023). As the results detailed in Appendix C.3, conventional MORL approaches struggled to identify non-dominated policies that align closely with specific DM preferences, especially in complex tasks like Humanoid-v2.

**Further algorithmic analysis**  Beyond these specific studies, we also conducted a comprehensive series of experiments to analyze the impact of PBMORL's other core algorithmic components and hyperparameters. These detailed findings are available in Appendices B.3 through B.7.

## 5. Related Works

**Single-policy MORL**  Single-policy MORL methods typically aggregate multiple objectives into a single scalar reward, guided by the DM's predefined preferences. Standard RL algorithms optimize this scalar reward to produce one optimal policy (Gábor et al., 1998; Mannor & Shimkin, 2001; Zhou et al., 2024). Aggregation can be linear or use more complex forms, such as exponential aggregation in (Rolf, 2020) and (Abdolmaleki et al., 2020) proposed learning different policies for each objective and then synthesizing them into a composite policy. Although conceptually simple, single-policy MORL approaches often assume DMs can accurately specify their preferences beforehand. This

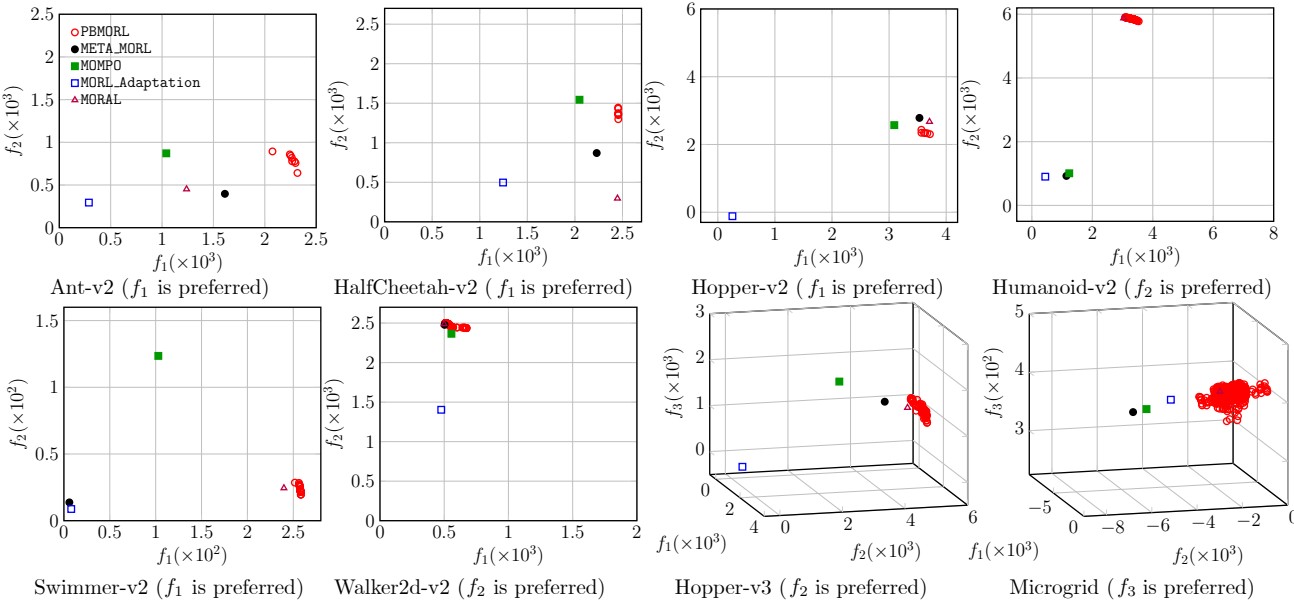

*Figure 3.* Selected plots of non-dominated policies obtained by `PBMORL` vs `MORL-Adaptation`, `META-MORL`, `MOMPO` and `MORAL`. Other results are provided in Fig. 10 of Appendix C.2.

*Table 2.* Comparison results of $\epsilon^{\star}(\Pi)$ of `PBMORL` versus peer algorithms over 10 runs with mean and standard deviation. The complete results for other six tasks are provided in Table 11 of Appendix C.2.

| | | PBMORL | MOMPO | META-MORL | MORL-Adaptation | MORAL |
|---|---|---|---|---|---|---|
| Humanoid-v2 | $f_1$ | **2.362(2.51E−2)** | 9.292(4.17E−1) | 8.679(5.18E−2) | 9.554(8.33E−2) | 5.599(6.40E−2) |
| | $f_2$ | **4.099(6.96E−7)** | 8.993(1.84E−2) | 9.083(2.84E−3) | 9.912(8.34E−2) | 9.776(1.34E−2) |
| MMSD | $f_1$ | **1.149(8.85E−4)** | 2.418(3.39E−3) | 2.737(9.05E−3) | 4.69(7.22E−3) | 2.48(6.72E−4) |
| | $f_2$ | **0.000(4.87E−7)** | 1.366(9.02E−4) | 1.317(8.62E−4) | 7.020(2.31E−4) | 0.672(5.56E−5) |
| | $f_3$ | **0.030(9.78E−8)** | 0.102(4.57E−6) | 0.071(8.76E−5) | 0.244(9.94E−4) | 0.124(1.29E−5) |

assumption rarely holds true in practice, limiting their usefulness.

**Multi-policy MORL** Multi-policy MORL decomposes the problem into multiple subproblems, each representing a distinct trade-off. Each subproblem is solved using single-objective RL algorithms. Some frameworks optimize these subproblems sequentially (Yang et al., 2019; Mossalam et al., 2016; Reymond & Nowé, 2019; Zhang et al., 2023; Mazouchi et al., 2022; He et al., 2025; Li et al., 2025). Others solve them concurrently, enabling parallel policy optimization (Xu et al., 2020; Parisi et al., 2014; Li et al., 2021; Shao et al., 2023). The key advantage of multi-policy approaches is the ability to find a diverse set of Pareto-optimal policies without requiring predefined preferences. However, they often incur significant computational costs. Additionally, presenting numerous policies can overwhelm DMs, especially when only a small subset is relevant.

**Preference learning** Preference-based MORL methods have gained recent attention. These approaches aim to learn a DM's utility function from interactive feedback, such as pairwise comparisons or rankings, to guide policy search (Roijers & Whiteson, 2017; Roijers et al., 2018; Wanigasekara et al., 2019). Many methods frame this as a multi-armed bandit problem. However, directly applying fixed-arm bandits to continuous, high-dimensional RL tasks can be challenging. Yang et al. (Yang et al., 2019) propose approximating the full PF first, then using a GP model to learn preferences afterward. Although effective, this approach can be computationally expensive if only a limited ROI is relevant. Additionally, policies outside the ROI may add noise to preference learning. Related methods include preference-based inverse RL, which infers reward functions from demonstrations (Sugiyama et al., 2012; Pan & Shen, 2018; Brown et al., 2019a; Lian et al., 2024; Perrusquía & Guo, 2025; Que et al., 2025). These methods require explicit demonstrations labeled by preferences, which can be difficult to provide comprehensively beforehand (Brown et al., 2019b). Some direct interaction methods modify policies based on DM feedback but may not integrate preferences

deeply into optimization (Kollmitz et al., 2020). Lastly, several methods learn parameterized representations of DM utilities to guide RL (Wirth et al., 2016; Christiano et al., 2017; Wu & Wang, 2024; Xiao et al., 2025; Wu et al., 2024; Xue et al., 2023). PBMORL builds upon these approaches but uniquely uses interactive preference elicitation. This interactive process dynamically biases a multi-policy search towards the ROI, avoiding the need to find the entire PF or rely solely on predefined demonstrations.

## 6. Conclusion

This paper proposed PBMORL, a human-in-the-loop framework for preference-based MORL that searches for policies of interest preferred by the DM. This framework proactively learns the DM's preferences in an interactive manner, using the learned preference information to guide policy optimization in MORL. Experiments on the quadruped robot control task, the MuJoCo benchmark and the MMSD task demonstrate the effectiveness of PBMORL for finding high-quality policies that align with the DM's preferences.

## Acknowledgments

This work was supported by the UKRI Future Leaders Fellowship under Grant MR/S017062/1 and MR/X011135/1; in part by National Natural Science Foundation of China under Grant 62376056 and 62076056; in part by the Science and Technology Innovation Program of Hunan Province under Grant 2024RC1011 and the Major Basic Research Projects in Hunan Province under Grant 2026JC0001; in part by the Isambard-AI, Royal Society Faraday Discovery Fellowship (FDF/S2/251014), BBSRC Transformative Research Technologies (UKRI1875), Royal Society International Exchanges Award (IES/R3/243136), Kan Tong Po Fellowship (KTP/R1/231017); and the Amazon Research Award and Alan Turing Fellowship.

## Impact Statement

This paper presents work whose goal is to advance the field of Machine Learning. There are many potential societal consequences of our work, none which we feel must be specifically highlighted here.

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

# A. Algorithmic Details

This section gives some technical details of our proposed PBMORL framework, including the method used to generate weight vectors and the relevant pseudo-codes.

## A.1. Weight vector generation

We employ the Das and Dennis's method (Das & Dennis, 1998) to generate a set of evenly distributed weight vectors along a unit simplex. The fundamental idea is to divide each coordinate into $H > 0$ equally spaced segments. Weight vectors are formed by iteratively selecting a sliced coordinate along each axis. The Das and Dennis's method generates a total of $\binom{H+m-1}{m-1}$ weight vectors. Each weight vector corresponds to a unique subproblem in PBMORL. Fig. 4 provides an illustrative example of 21 weight vectors generated in a 3-dimensional space with $H = 5$.

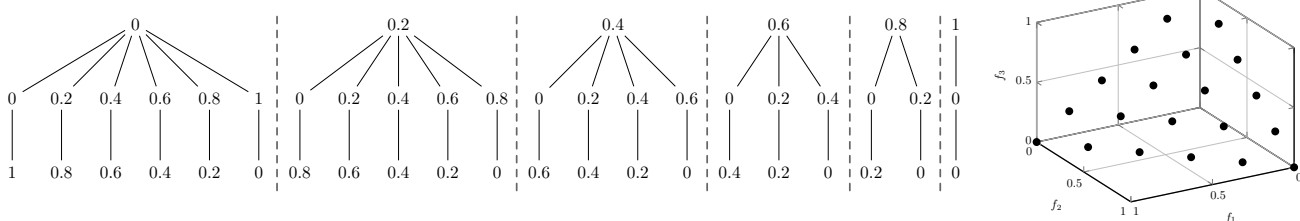

*Figure 4.* An illustrative example of weight vector generation in the three-dimensional space by using the Das and Dennis's method.

## A.2. Pseudocode

The pseudo codes of PBMORL, the preference learning process, and the MOPPO are given in Algorithm 1, Algorithm 2, and Algorithm 3, respectively.

---

**Algorithm 1** Pseudo code of PBMORL

---

1: **Input:**
2:     Number of total environment steps: $st_{env}$,
3:     Number of environment steps for seeding module: $st_{sd}$,
4:     Number of environment steps in one round of MORL: $st_r$,
5:     Number of interactions with the DM: $q$,
6:     Number of subtasks: $n_w$
7: Generate weight vectors $\mathcal{W}_{ini} := \{\mathbf{w}^i\}_{i=1}^{n_w}$ using Das and Dennis' method
8: Initialize $\Pi$ for storing policies
9: Initialize EP for storing the non-dominated policies
10: Initialize $\{\pi^i\}_{i=1}^{n_w}$, and assign each one with a weight vector in $\mathcal{W}_{ini}$ to constitute $\Pi := \{\langle \pi^i, \mathbf{w}^i \rangle\}_{i=1}^{n_w}$
11: Initialize preference model $u(\pi)$, and dataset $\tilde{\Pi}$, which stores the training data for preference learning
12: Get promising policies from the seeding module
13: **while** $st_{sd}$ is not met **do**
14:     $\Pi \leftarrow \text{MOPPO}(st_r, \Pi)$
15: **end while**
16: Get non-dominated policies following preference elicitation module and policy optimization module
17: **while** $st_{env}$ is not met **do**
18:     **if** query the DM and $q$ is not met **then**
19:         $\Pi, \tilde{\Pi}, u(\pi) \leftarrow \text{Preference Learning}(\Pi, \tilde{\Pi}, u(\pi))$
20:     **end if**
21:     $\Pi \leftarrow \text{MOPPO}(st_r, \Pi)$
22:     Update EP with non-dominated policies of $\Pi$
23: **end while**
24: **Return** EP

---

---

**Algorithm 2** `preference elicitation`

---

1: **Input:** $\Pi$: stored non-dominated policies, $\tilde{\Pi}$: queried policies as training data, $u(\pi)$: preference model
2: **Consultation step:**
3: **if** random selection **then**
4:    Randomly sample two policies $\pi^1$, $\pi^2$ from $\Pi$.
5: **else**
6:    **for all** $\pi \in \Pi$ **do**
7:       Evaluate its utility value $u(\pi)$ and $\mathtt{I}(\pi)$.
8:    **end for**
9:    $\pi^1 := \arg\max_{\pi \in \Pi} \mathtt{I}(\pi)$
10:   $\pi^2 := \arg\max_{\pi \in \Pi \setminus \{\pi^1\}} \mathtt{I}(\pi)$
11: **end if**
12: Query the DM with $\pi^1$ and $\pi^2$, and store the result in $\tilde{\Pi}$.
13: **Preference learning step:**
14: Use the DM's feedback to update $\tilde{\Pi}$ and the preference model $u(\pi)$ as in Section 3.2.2.
15: **Preference translation step:**
16: Generate the new policy set $\Pi$ as in Section 3.2.3.
17: **Return** $\Pi$, $\tilde{\Pi}$, $u(\pi)$.

---

**Algorithm 3** `MOPPO`

---

1: **Input:** $t_{\mathrm{r}}$: # of environment steps, $\overline{\Pi}$: stored policies
2: Initialize $\Pi := \emptyset$ for storing optimized policies.
3: **for all** $\langle \pi^i, \mathbf{w}^i \rangle \in \overline{\Pi}$ **do**
4:    **while** the stopping criterion is not met **do**
5:       Execute $\pi^i$ and store collected trajectory data in buffer:
6:       $\mathcal{S}^i := \left( \langle s_1^i, a_1^i, r_1^i \rangle, \ldots, \langle s_{t_{\mathrm{r}}}^i, a_{t_{\mathrm{r}}}^i, r_{t_{\mathrm{r}}}^i \rangle \right)$.
7:       **while** the stopping criterion is not met **do**
8:          Sample a trajectory from $\mathcal{S}^i$, and use $\mathbf{w}^i$ to aggregate its rewards.
9:          Compute $\hat{\mathcal{A}}_t$ as in (Schulman et al., 2016).
10:         Update $\pi^i$ by optimizing $J^{\mathrm{clip}}(\pi^i)$.
11:      **end while**
12:      Insert updated task $\langle \pi^i, \mathbf{w}^i \rangle$ to $\Pi$.
13:   **end while**
14: **end for**
15: **Return** Non-dominated policies in $\Pi$.

---

## B. Ablation Study and Further Analyses

The previous experiments demonstrated the effectiveness of `PBMORL` for finding the policies of interest according to the DM's preferences. In this section, we further analyze several core algorithmic components by addressing the following research questions (RQs).

RQ1: What is impact of the interaction frequency?

RQ2: What is the benefit of our GP-based `preference learning` module?

RQ3: When do we consult the DM?

RQ4: What is the impact of the parameters $\kappa_1$ and $\kappa_2$ in the `preference translation` step?

RQ5: What if the DM's preference is not determinstic?

RQ6: What if we use other the aggregation function other than the linear aggregation in equation (2)?

*Table 3.* Comparison results of PBMORL with different numbers of interactions.

| | | PBMORL(40 queries) | PBMORL(20 queries) | PBMORL(10 queries) | MOMPO | META-MORL | MORL-Adaptation | MORAL |
|---|---|---|---|---|---|---|---|---|
| Ant-v2 | $f_1$ | **7.709** | **7.710** | 7.833 | 8.910 | 8.355 | 9.686 | 8.926 |
| | $f_2$ | **7.637** | 7.690 | 7.690 | 8.914 | 8.446 | 9.633 | 8.377 |
| HalfCheetah-v2 | $f_1$ | **7.541** | **7.541** | **7.543** | 7.953 | 7.752 | 8.743 | 7.545 |
| | $f_2$ | **7.500** | **7.498** | 7.582 | 7.500 | 7.980 | 8.561 | 7.500 |
| Hopper-v2 | $f_1$ | **6.221** | 6.405 | 6.527 | 6.828 | 6.404 | 9.662 | 6.236 |
| | $f_2$ | **4.587** | 4.601 | 5.355 | 8.583 | 4.930 | 9.942 | 9.761 |
| Humanoid-v2 | $f_1$ | **2.362** | 5.164 | 5.773 | 9.292 | 8.679 | 9.554 | 5.599 |
| | $f_2$ | **4.099** | 4.495 | 5.173 | 8.993 | 9.083 | 9.912 | 9.776 |
| Swimmer-v2 | $f_1$ | **9.747** | **9.749** | **9.747** | 9.913 | 9.933 | 9.941 | 9.759 |
| | $f_2$ | **9.850** | **9.850** | **9.850** | 9.850 | 9.860 | 9.850 | 9.850 |
| Walker2d-v2 | $f_1$ | **8.048** | 8.337 | 8.502 | 8.905 | 8.267 | 9.271 | 8.850 |
| | $f_2$ | **7.500** | 7.699 | 9.541 | 7.643 | 7.528 | 8.602 | 7.500 |
| Hopper-v3 | $f_1$ | **6.020** | 6.081 | 6.101 | 6.821 | 6.773 | 9.070 | 6.391 |
| | $f_2$ | **4.346** | 4.833 | 5.796 | 7.295 | 6.032 | 9.411 | 5.263 |
| | $f_3$ | **7.500** | **7.500** | **7.500** | 7.500 | 7.522 | 7.500 | 7.500 |
| MMSD | $f_1$ | **1.149** | **1.149** | **1.149** | 2.418 | 2.737 | 4.69 | 2.48 |
| | $f_2$ | **0.000** | **0.000** | **0.000** | 1.366 | 1.317 | 7.020 | 0.672 |
| | $f_3$ | **0.030** | **0.031** | **0.031** | 0.102 | 0.071 | 0.244 | 0.124 |

RQ7: What is the impact of the duration of the seeding phase?

RQ8: What is the impact of the parameters $\alpha$, $\beta$ and $\eta$?

## B.1. PBMORL efficiently identifies human preferences

In the consultation stage, more frequent interactions with the DM would generate more oracles for preference learning. However, this also significantly increases the DM's workload. To address RQ2, we set the number of interactions to 20 and compared this with 40 and 10 consultations. The results are shown in Fig. 5 and Table 3. As seen in Table 3, PBMORL achieves strong performance even with as few as 10 to 20 interactions. Specifically, PBMORL with only 10 interactions still outperforms MORAL on most tasks. We observe that reducing the number of interactions does not negatively impact PBMORL's performance on some problems like HalfCheetah-v2 with a preference for $f_2$. However, for other problems like Walker2d-v2 with a preference for $f_2$, decreasing the number of queries to 10 can significantly hurt PBMORL's performance. On the other hand, increasing the number of interactions does not bring significantly better performance but can increase the DM's cognitive load.

## B.2. GP-based preference learning enables superior performance

In principle, any off-the-shelf preference learning methods can be used for modeling the DM's preference information. To validate this assertion, we construct two variants of PBMORL, dubbed RSMORL and WLMORL. They replace the preference learning method introduced in Section 3.2.2 with a classic preference learning approach ranking-SVM (Joachims, 2002) and a preference learning method in RL (Wirth et al., 2016), respectively. Note that all variants apply one query at each round of policy optimization, as done in PBMORL. The comparison results shown in Table 4 demonstrate the better performance of PBMORL on most problems. Specifically, comparing to RSMORL, our better performance not only highlights the superiority of GP for preference learning, but also showcases the benefits of uncertainty provided by GP prediction for better exploration. Comparing to WLMORL as well as MORAL, our better performance demonstrates that learning preferred weight vector(s) is less reliable than learning the DM's preference in the objective space.

## B.3. A priori preferences are difficult to specify correctly

In addition to the *interactive* preference elicitation considered in PBMORL, DM's preferences can also be incorporated in *a priori* or *a posteriori* manner. From the comparison results discussed in Appendix C.3, we can see that the a posteriori

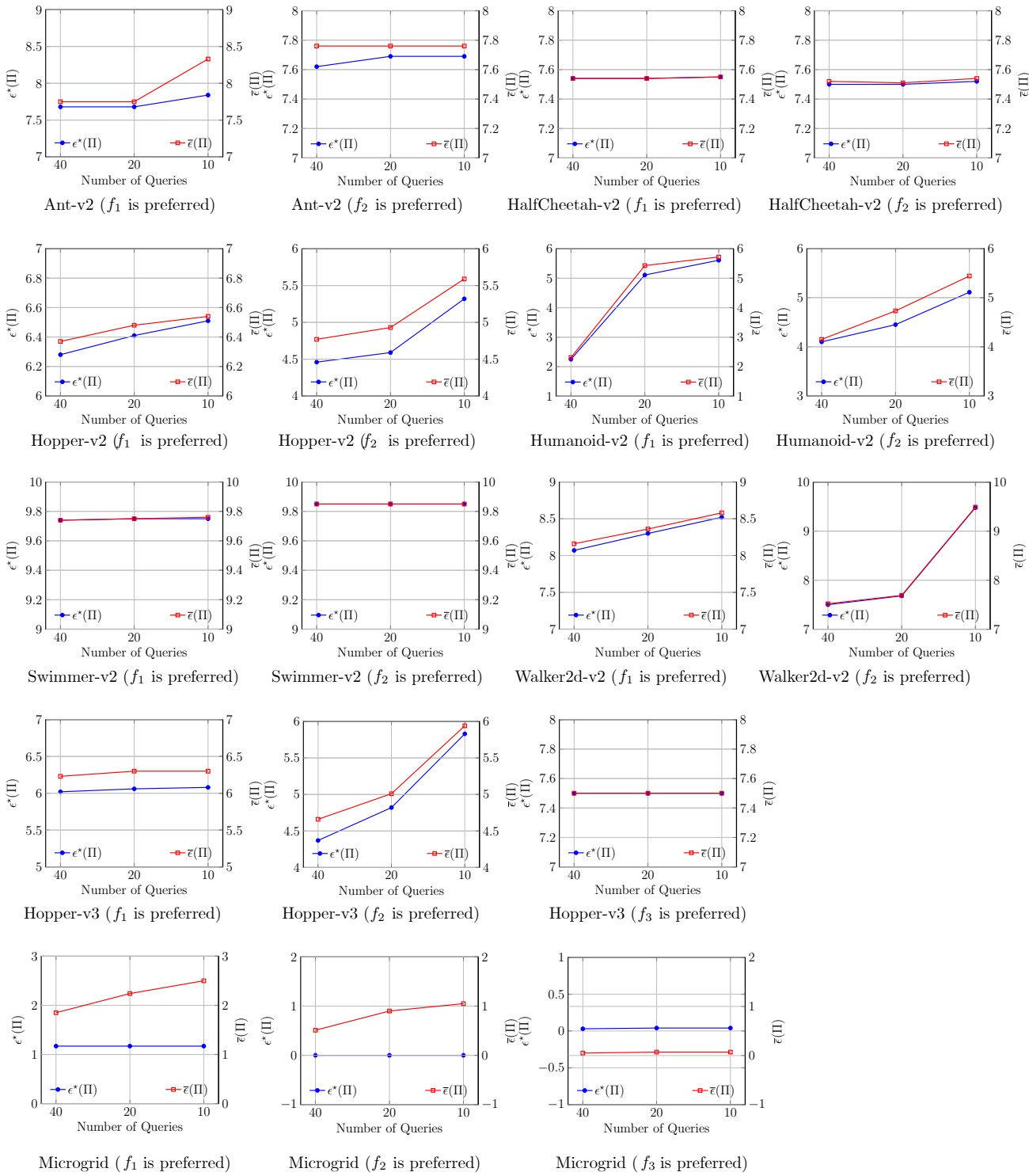

*Figure 5.* Comparison results of $\epsilon^\star(\Pi)$ and $\bar{\epsilon}(\Pi)$ obtained by PBMORL with 10, 20, and 40 interactions, respectively.

method may fail to identify the policy of interest in some of the challenging problems. The better performance of PBMORL against the other four peer preference-based MORL, as shown in Section 4.2, demonstrate that the interactive MORL outperforms the a priori method. To have a better understanding of the discrepancy between the a priori and interactive preference elicitation, we plot the corresponding weight vector a priori specified by the DM versus those identified by

*Table 4.* Comparison results of $\epsilon^\star(\Pi)$ and $\overline{\epsilon}(\Pi)$ of PBMORL versus RSMORL and WLMORL over 10 runs with mean and standard deviation.

| | | PBMORL | | RSMORL | | WLMORL | |
|---|---|---|---|---|---|---|---|
| | | $\epsilon^\star(\Pi)$ | $\overline{\epsilon}(\Pi)$ | $\epsilon^\star(\Pi)$ | $\overline{\epsilon}(\Pi)$ | $\epsilon^\star(\Pi)$ | $\overline{\epsilon}(\Pi)$ |
| Ant-v2 | $f_1$ | **7.709(1.68E−3)** | **7.880(3.32E−2)** | 8.333(1.44E−5) | 8.796(6.22E−3) | 7.746(1.35E−5) | 7.949(5.02E−3) |
| | $f_2$ | **7.637(6.98E−4)** | **7.751(1.65E−4)** | 7.722(1.24E−5) | 7.998(5.72E−3) | 7.695(8.74E−7) | 7.956(2.23E−4) |
| HalfCheetah-v2 | $f_1$ | **7.541(3.21E−6)** | **7.542(7.53E−6)** | 7.566(3.71E−4) | 7.721(5.29E−4) | 8.397(1.31E−4) | 9.233(8.53E−2) |
| | $f_2$ | **7.500(3.70E−9)** | **7.508(1.17E−4)** | **7.500(2.47E−7)** | 7.541(2.14E−6) | **7.508(7.98E−3)** | 7.544(8.98E−2) |
| Hopper-v2 | $f_1$ | **6.221(1.14E−2)** | **6.299(1.57E−2)** | 9.566(8.17E−5) | 9.642(5.58E−2) | 9.421(6.43E−7) | 9.631(5.16E−3) |
| | $f_2$ | **4.587(2.25E−2)** | **4.972(3.11E−2)** | 8.351(5.29E−5) | 8.635(3.68E−2) | 9.922(3.11E−5) | 9.730(6.21E−3) |
| Humanoid-v2 | $f_1$ | **2.362(2.51E−2)** | **2.583(8.31E−1)** | 6.324(5.21E−5) | 6.543(5.19E−3) | 9.236(5.66E−5) | 9.353(6.74E−2) |
| | $f_2$ | **4.099(6.96E−7)** | **4.143(9.79E−5)** | 4.818(5.89E−5) | 5.076(2.29E−2) | 8.718(1.39E−6) | 8.728(5.21E−2) |
| Swimmer-v2 | $f_1$ | **9.747(1.12E−4)** | **9.749(1.49E−4)** | 9.716(6.23E−5) | 9.762(7.22E−3) | 9.884(7.22E−3) | 9.894(2.39E−3) |
| | $f_2$ | **9.850(1.00E−12)** | **9.850(3.67E−12)** | **9.850(5.39E−8)** | **9.850(1.12E−7)** | **9.850(8.84E−7)** | **9.850(6.23E−6)** |
| Walker2d-v2 | $f_1$ | **8.048(9.77E−4)** | **8.116(3.95E−3)** | 8.636(5.13E−5) | 8.928(7.11E−2) | 8.609(4.26E−5) | 8.897(9.21E−3) |
| | $f_2$ | **7.500(3.21E−8)** | **7.506(8.84E−5)** | **7.500(6.64E−6)** | 7.534(8.34E−4) | 7.600(1.14E−5) | 7.673(9.96E−3) |
| Hopper-v3 | $f_1$ | 6.020(8.91E−8) | **6.203(1.44E−3)** | **5.981(5.12E−5)** | 6.250(9.75E−3) | 6.858(6.75E−6) | 7.736(7.34E−3) |
| | $f_2$ | **4.346(2.02E−3)** | **4.539(1.97E−2)** | 5.000(2.66E−5) | 5.272(9.74E−4) | 8.611(1.74E−6) | 9.657(8.72E−4) |
| | $f_3$ | **7.500(1.90E−8)** | **7.502(1.23E−5)** | **7.500(2.74E−4)** | 7.510(8.82E−3) | **7.500(1.08E−6)** | **7.500(4.34E−3)** |
| MMSD | $f_1$ | **1.149(8.85E−4)** | **1.846(3.88E−5)** | **1.149(1.34E−3)** | 2.382(8.87E−2) | **1.149(5.88E−5)** | 2.397(1.74E−3) |
| | $f_2$ | **0.000(4.87E−7)** | **0.511(3.41E−6)** | **0.000(2.62E−5)** | 1.558(6.62E−3) | **0.000(2.34E−5)** | 1.045(8.78E−4) |
| | $f_3$ | **0.030(9.78E−8)** | **0.050(5.74E−8)** | **0.030(1.22E−5)** | 0.081(8.87E−7) | **0.030(3.54E−6)** | 0.074(3.79E−4) |

*Table 5.* Comparison results of $\epsilon^\star(\Pi)$ and $\overline{\epsilon}(\Pi)$ on different settings of $\kappa_1$ and $\kappa_2$ over 10 runs with mean and standard deviation.

| | | $\kappa_1 = 100\% \times |\Pi|, \kappa_2 = 0$ | | $\kappa_1 = 80\% \times |\Pi|, \kappa_2 = 20\% \times |\Pi|$ | | $\kappa_1 = 50\% \times |\Pi|, \kappa_2 = 50\% \times |\Pi|$ | | $\kappa_1 = 20\% \times |\Pi|, \kappa_2 = 80\% \times |\Pi|$ | |
|---|---|---|---|---|---|---|---|---|---|
| | | $\epsilon^\star(\Pi)$ | $\overline{\epsilon}(\Pi)$ | $\epsilon^\star(\Pi)$ | $\overline{\epsilon}(\Pi)$ | $\epsilon^\star(\Pi)$ | $\overline{\epsilon}(\Pi)$ | $\epsilon^\star(\Pi)$ | $\overline{\epsilon}(\Pi)$ |
| HalfCheetah-v2 | $f_1$ | 7.541(7.65E−8) | 7.542(6.34E−7) | **7.541(3.21E−6)** | **7.542(7.53E−6)** | 7.550(6.24*E*-8) | 7.554(7.78E−7) | 7.551(8.75E−8) | 7.562(2.56E−7) |
| | $f_2$ | **7.500(6.76E−10)** | **7.508(3.19E−9)** | **7.500(3.70E−9)** | **7.508(1.17E−8)** | **7.500(4.42E−9)** | 7.522(8.12E−4) | **7.500(6.24E−6)** | 7.534(3.28E−6) |
| Swimmer-v2 | $f_1$ | 9.761(6.79E−5) | 9.762(3.78E−4) | **9.747(1.12E−4)** | **9.749(1.49E−4)** | 9.770(5.63E−4) | 9.773(3.29E−5) | 9.763(6.79E−5) | 9.768(1.12E−5) |
| | $f_2$ | **9.850(4.55E−10)** | **9.850(7.87E−11)** | **9.850(1.00E−12)** | **9.850(3.67E−12)** | **9.850(2.56E−10)** | **9.850(7.73E−10)** | **9.850(6.66E−9)** | **9.850(8.34E−9)** |
| Walker2d-v2 | $f_1$ | 8.612(2.37E−4) | 8.652(6.17E−4) | **8.048(9.77E−4)** | **8.116(3.95E−3)** | 8.623(8.13E−3) | 8.738(6.32E−2) | 8.513(7.93E−4) | 8.671(5.35E−3) |
| | $f_2$ | **7.500(6.61E−6)** | 7.593(6.25E−5) | **7.500(3.21E−8)** | **7.506(8.84E−5)** | **7.500(7.27E−6)** | 7.510(3.90E−4) | **7.500(1.22E−5)** | 7.516(8.28E−4) |

PBMORL, as shown in Fig. 6. From these plots, we can see that the weight vector specified by the DM is always outside the region of interest (ROI). From the DM's perspective, it is not intuitive for the DM to elicit an appropriate weight vector a priori given the black-box nature of the problem itself. Our experiments demonstrate that there is even no guarantee to find good non-dominated policies without considering the DM's preference information before a posteriori decision-making.

## B.4. Analysis of $\kappa_1$ and $\kappa_2$

In the `preference translation` step, there are two hyperparameters $\kappa_1$ and $\kappa_2$ that implicitly control the balance between exploration versus exploitation. Specifically, a larger $\kappa_1$ indicates a greater reliance on learned preference information to guide policy optimization, while a larger $\kappa_2$ emphasizes random exploration. We set $\kappa_1 = 80\% \times |\Pi|$ and $\kappa_2 = 20\% \times |\Pi|$ as the default. To address RQ4, here we empirically compare the the default setting with three other $\kappa_1 \in \{100\% \times |\Pi|, 50\% \times |\Pi|, 20\% \times |\Pi|\}$ on three example problems, including Walker2d-v2, HalfCheetah-v2, and Swimmer-v2. From the comparison results in Table 5, we find that the setting with $\kappa_1 = 80\% \times |\Pi|$ and $\kappa_2 = 20\% \times |\Pi|$ consistently achieves the best performance when considering $\epsilon^\star(\Pi)$. On the other hand, when considering $\overline{\epsilon}(\Pi)$, the outcomes depend more on the characteristics of the individual problems. In general, we find that a large $\kappa_1$ and a nonzero $\kappa_2$ are efficient for most situations.

## B.5. PBMORL effectively handles fuzzy preferences

The DM's preferences discussed so far in our experiments are all *deterministic*. That is to say DMs are assumed to prefer only one objective function over the other(s). However, it is not uncommon that DMs can be blurry about their preference. For instance, the DM can be more into one objective (say 70%), but she also gives certain level of priority (say the remaining

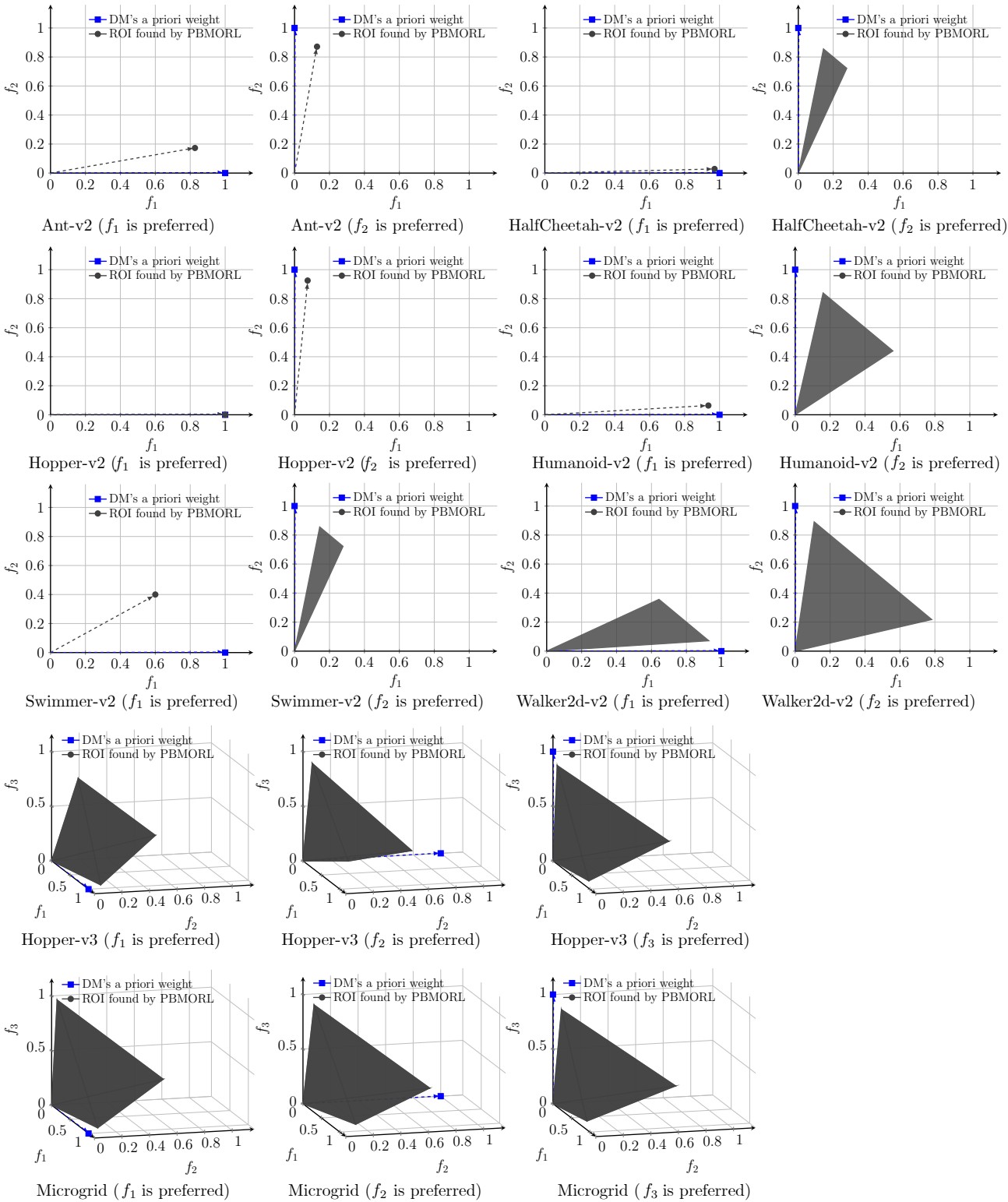

*Figure 6.* Comparison of the weight vector specified by the DM *a priori* versus the ROI identified by `PBMORL` (shaded in the gray region).

30%) to the other objective(s). Here, we conduct an experiment that considers a *fuzzy* type of preference. From the plots of the non-dominated policies found by different types of preference settings in Fig. 7, we find that our proposed `PBMORL` can

also be used to find trade-off policies with a controllable bias towards one of the objectives, instead of a polarized preference. However, we also find that the trade-off policies found by PBMORL in Ant-v2 are further polarized in the less preferred objective. This can be attributed to the intriguing interaction of two objectives in Ant-v2 or the difficulty of PBMORL when tackling this environment. All in all, it is an interesting and important future direction to investigate more diversified types of preference elicitation methods under the PBMORL framework.

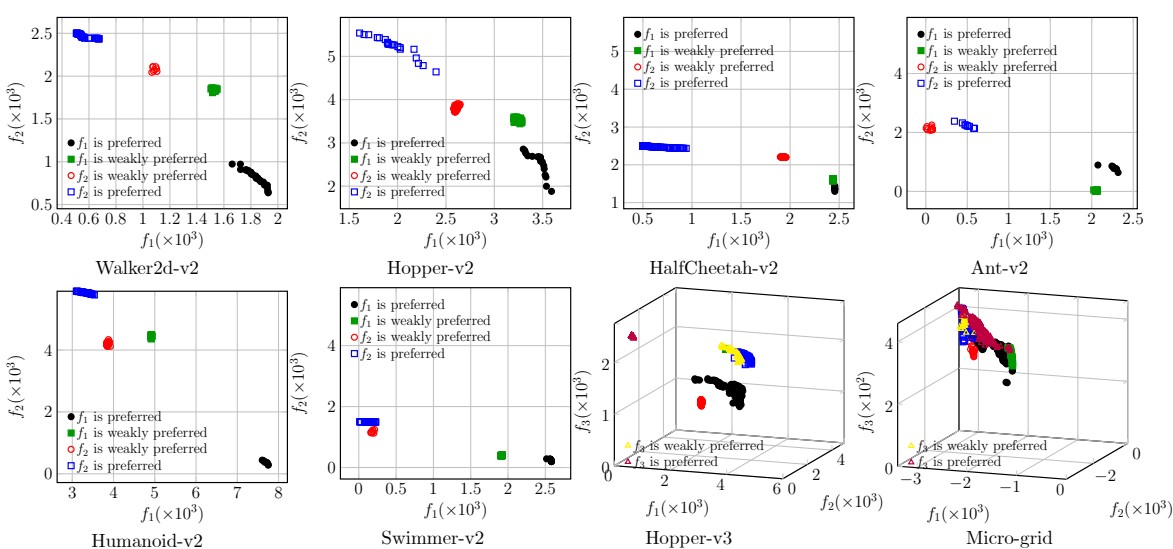

*Figure 7.* Selected plots of the non-dominated policies obtained by PBMORL with different types of preference settings. One is *deterministic* preference on one objective while the other is a *fuzzy* type of preference.

### B.6. PBMORL remains robust under different aggregation functions

As one of the key components of PBMORL, the seeding module works as a conventional MORL to search for a set of promising trade-off policies that approximate the PF. While we applied a linear aggregation in PBMORL for a proof-of-concept purpose, it has been notorious in the multi-objective optimization domain (e.g., (Deb, 2001; Zhang & Li, 2007; Li et al., 2021)) that linear aggregation can be ineffective to search for solutions located in the non-convex region(s) of the PF. In contrast, the following weighted Tchebycheff aggregation function has been widely recognized to be applicable to problems with both convex and non-convex regions in the PF:

$$\tilde{J}(\pi, \mathbf{w}, \mathbf{z}) = \max_{i \in \{1, \ldots, m\}} \left\{ w_i | J_i(\pi) - z_i | \right\}, \tag{12}$$

where $\mathbf{z} = (z_1, \cdots, z_m)^\top$ is the utopia point. Note that all objective functions in this paper are considered being maximized, whereas there is no *a priori* knowledge about the maximum of each objective function. In this case, we set $\mathbf{z}$ as the nadir point instead, i.e., $z_i = 0, i \in \{1, \ldots, m\}$. To investigate RQ6, we use equation (12) in the seeding module of PBMORL to constitute a variant dubbed PBMORL-TCH. Note that our proposed PBMORL is a general framework that each component can be adapted to any other techniques in a plug-in manner. From the comparison results of $\epsilon^\star(\Pi)$ and $\bar{\epsilon}(\Pi)$ shown in Table 6 along with the non-dominated policies shown in Fig. 8, we can see that the performance of PBMORL and PBMORL-TCH is close to each other. This can be explained as the PF of the MORL problems considered here are all with convex PFs. The robust performance PBMORL-TCH also provides us confidence to extend our proposed PBMORL framework for handling problems in more complex environments.

### B.7. Analysis of $\alpha$, $\beta$ and $\eta$

We conduct sensitivity analyses for $\alpha$ in equation (3) and $\beta$ in Section 3.2.3, with the results summarized in Table 7 and Table 8.

The parameter $\alpha$ controls the balance between exploration and exploitation during preference consultation. A larger $\alpha$ encourages querying policies with higher uncertainty. In our main experiments, we set $\alpha = 1.0$, and additionally test

*Table 6.* Comparison results of $\epsilon^\star(\Pi)$ and $\bar{\epsilon}(\Pi)$ of PBMORL against PBMORL-TCH over 10 runs with mean and standard deviation.

| | | PBMORL-TCH | | PBMORL | |
|---|---|---|---|---|---|
| | | $\epsilon^\star(\Pi)$ | $\bar{\epsilon}(\Pi)$ | $\epsilon^\star(\Pi)$ | $\bar{\epsilon}(\Pi)$ |
| Ant-v2 | $f_1$ | **7.738(6.41E−3)** | **7.866(5.09E−2)** | 7.709(1.68E−3) | 7.880(3.32E−2) |
| | $f_2$ | 7.700(3.70E−3) | 7.87(2.06E−2) | **7.637(6.98E−4)** | **7.751(1.65E−4)** |
| HalfCheetah-v2 | $f_1$ | 7.543(2.00E−6) | 7.591(1.06E−2) | **7.541(3.21E−6)** | **7.542(7.53E−6)** |
| | $f_2$ | 7.501(3.23E−7) | 7.510(2.63E−5) | **7.500(3.70E−9)** | **7.508(1.17E−4)** |
| Hopper-v2 | $f_1$ | **6.203(3.71E−3)** | 6.301(1.13E−2) | 6.221(1.14E−2) | **6.299(1.57E−2)** |
| | $f_2$ | 4.700(4.34E−2) | **4.904(1.04E−1)** | **4.587(2.25E−2)** | 4.972(3.11E−2) |
| Humanoid-v2 | $f_1$ | **2.230(3.64E−4)** | **2.289(3.39E−5)** | 2.362(2.51E−2) | 2.583(1.49E−1) |
| | $f_2$ | 4.224(3.55E−3) | 4.284(2.72E−3) | **4.099(6.96E−7)** | **4.143(9.79E−5)** |
| Swimmer-v2 | $f_1$ | 9.751(7.69E−5) | 9.752(1.04E−4) | **9.747(1.12E−4)** | **9.749(1.49E−4)** |
| | $f_2$ | **9.850(3.13E−12)** | **9.850(1.04E−7)** | **9.850(1.00E−12)** | **9.850(3.67E−12)** |
| Walker2d-v2 | $f_1$ | **7.922(1.76E−1)** | **8.067(1.86E−1)** | 8.048(9.77E−4) | 8.116(3.95E−3) |
| | $f_2$ | **7.500(4.74E−6)** | 7.525(3.65E−4) | **7.500(3.21E−8)** | **7.506(8.84E−5)** |
| Hopper-v3 | $f_1$ | 6.360(5.29E−2) | 6.582(6.21E−2) | **6.020(8.91E−8)** | **6.203(1.44E−3)** |
| | $f_2$ | **4.333(1.58E−1)** | **4.434(4.21E−4)** | 4.346(2.02E−3) | 4.539(1.97E−2) |
| | $f_3$ | **7.500(5.60E−6)** | 7.506(2.05E−5) | **7.500(1.90E−8)** | **7.502(1.23E−5)** |
| MMSD | $f_1$ | 1.169(3.96E−1) | **1.206(5.20E−4)** | **1.149(8.85E−4)** | 1.846(3.88E−5) |
| | $f_2$ | 0.016(4.94E−4) | **0.086(1.13E−2)** | **0.000(4.87E−7)** | 0.511(3.41E−6) |
| | $f_3$ | 0.039(5.00E−7) | **0.047(2.45E−5)** | **0.030(9.78E−8)** | 0.050(5.74E−8) |

*Figure 8.* Plots of the non-dominated policies obtained by PBMORL versus PBMORL-TCH with different preferences on each objective

$\alpha \in \{0.1, 2.0\}$. As shown in Table 7, $\alpha = 1.0$ yields the best trade-off across all three preference settings, while $\alpha = 0.1$ (insufficient exploration) leads to significantly worse performance.

The parameter $\beta$ balances exploitation and uncertainty when ranking policies. In our main experiments, we set $\beta = 0.1$, and additionally test $\beta \in \{0.05, 0.5\}$. The results in Table 8 demonstrate that $\beta = 0.1$ consistently achieves the best performance across all preference settings.

In addition, we analyze the impact of the weight bias parameter $\eta$. As summarized in Table 9, our chosen value of $\eta = 0.2$ provides a robust balance between biasing the search towards the region of interest and allowing sufficient exploration of new solutions. When $\eta = 0.5$, overall performance slightly declines, while setting $\eta = 1.0$ removes exploration entirely, leading to a significant degradation in policy quality.

*Table 7.* Sensitivity analysis for $\alpha$.

| Preference | $\alpha$ | Average Speed (m/s) ↑ | Average Torque (N·m) ↓ |
|---|---|---|---|
| Stationary | 0.1 | 0.006 | 10.468 |
| | 1.0 | **0.004** | **6.275** |
| | 2.0 | 0.004 | 7.193 |
| Moderate speed | 0.1 | 3.855 | 39.302 |
| | 1.0 | **2.600** | **13.682** |
| | 2.0 | 3.302 | 20.739 |
| High speed | 0.1 | 4.379 | 41.468 |
| | 1.0 | **5.544** | **28.463** |
| | 2.0 | 4.882 | 40.971 |

*Table 8.* Sensitivity analysis for $\beta$.

| Preference | $\beta$ | Average Speed (m/s) ↑ | Average Torque (N·m) ↓ |
|---|---|---|---|
| Stationary | 0.05 | 0.007 | 6.936 |
| | 0.1 | **0.004** | **6.275** |
| | 0.5 | 0.005 | 6.493 |
| Moderate speed | 0.05 | 3.152 | 30.688 |
| | 0.1 | **2.600** | **13.682** |
| | 0.5 | 2.953 | 17.300 |
| High speed | 0.05 | 5.390 | 28.922 |
| | 0.1 | **5.544** | **28.463** |
| | 0.5 | 4.910 | 27.495 |

## B.8. Impact of the duration of the seeding phase

In the early stages of training, policies perform very poorly, and interacting with the DM using such low-quality policies is inefficient. Thus, we introduce a seeding module that initially trains with a fixed set of preference weights to obtain a set of moderately performing policies, which then serve as the foundation for further preference learning. We investigate the impact of the seeding module's duration on the final performance across three different MuJoCo tasks. The results are presented in Fig. 9. They show that in simpler environments (e.g., Swimmer-v2) the seeding module has minimal effect, whereas in more complex settings (e.g., Humanoid-v2) its influence is more significant. This may be because in simpler tasks, acceptable policies can be learned quickly even without a dedicated seeding stage.

## C. Additional Results

### C.1. PBMORL scales effectively to many-objective settings

Most existing MORL algorithms only consider 2–3 objectives, with only a few toy-level environments (Reymond et al., 2022) involving more. However, we believe that it is important to examine the scalability of PBMORL in higher-dimensional objective spaces. To this end, we design and run a 5-objective experiment on the challenging Unitree Go2 task. The five objectives are:

- **Objective 1 (maximize forward speed)**: $r_{\text{lin\_vel}} = v_{\text{lin\_vel}}$, which encourages higher forward speed.

- **Objective 2 (minimize joint torque)**: $r_{\text{torq}} = -C_\tau \sum_{i=1}^{12} \tau_i^2 + 0.1$, which penalizes high joint torques to reduce energy consumption and wear on the actuators.

- **Objective 3 (minimize vertical velocity)**: $r_{\text{lin\_vel\_z}} = -C_z v_{\text{lin\_vel\_z}} + 0.1$, which penalizes the robot's vertical (Z-axis) velocity to encourage stable planar movement.

- **Objective 4 (minimize horizontal angular velocity)**: $r_{\text{ang\_vel\_xy}} = -C_\omega \omega_{\text{ang\_vel\_xy}} + 0.1$, which penalizes angular

*Table 9.* Sensitivity analysis for $\eta$.

| Preference | $\eta$ | Average Speed (m/s) $\uparrow$ | Average Torque (N·m) $\downarrow$ |
|---|---|---|---|
| Stationary | 0.2 | **0.004** | **6.275** |
| | 0.5 | 0.004 | 6.997 |
| | 1.0 | 0.011 | 10.507 |
| Moderate speed | 0.2 | **2.600** | **13.682** |
| | 0.5 | 2.309 | 13.563 |
| | 1.0 | 3.876 | 29.440 |
| High speed | 0.2 | **5.544** | **28.463** |
| | 0.5 | 4.833 | 27.540 |
| | 1.0 | 5.331 | 39.402 |

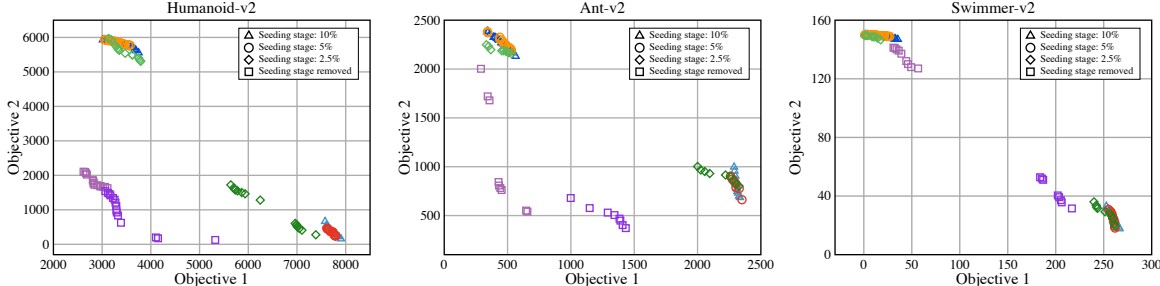

*Figure 9.* Impact of the duration of the seeding phase on Humanoid-v2, Ant-v2, and Swimmer-v2.

velocity in the horizontal (X and Y) axes to promote smoother and more controlled rotations.

- **Objective 5 (minimize action rate)**: $r_{\text{action\_rate}} = -C_a \sum (a_{t-1} - a_t)^2 + 0.1$, which penalizes large changes in action commands between consecutive steps, fostering smoother and more consistent control actions.

The scaling factors in the reward functions are used to bring different reward signals to a comparable magnitude, where $C_\tau = 5 \times 10^{-3}$, $C_z = 5$, $C_\omega = 0.15$, and $C_a = 0.01$.

As shown in Table 10, PBMORL successfully identifies high-quality policies aligned with human preferences even in the 5-objective environment. Compared to the original 2-objective results in Table 1, its performance remains remarkably strong, exhibiting only a minor drop despite the increased complexity of optimizing in a 5-objective space.

*Table 10.* Results in the 5-objective scenario.

| Preference | Reward Vector | Average Speed (m/s) $\uparrow$ | Average Torque (N·m) $\downarrow$ |
|---|---|---|---|
| Stationary | $[0.10, 98.0, 98.8, 99.4, 99.9]$ | 0.005 | 6.928 |
| Moderate speed | $[62.6, 82.8, 86.7, 60.4, 81.9]$ | 3.130 | 20.317 |
| High speed | $[108.2, 53.2, 39.6, 43.0, 44.8]$ | 5.410 | 33.514 |

### C.2. Comparison with preference-based MORL

This section presents the complete results of `PBMORL` compared with peer algorithms, including `MORL-Adaptation`, `META-MORL`, `MOMPO`, and `MORAL`. Fig. 10 illustrates the non-dominated policies obtained by different algorithms, while Table 11 summarizes the results of $\epsilon^\star(\Pi)$ for `PBMORL` versus peer algorithms over 10 runs, reporting both the mean and standard deviation.

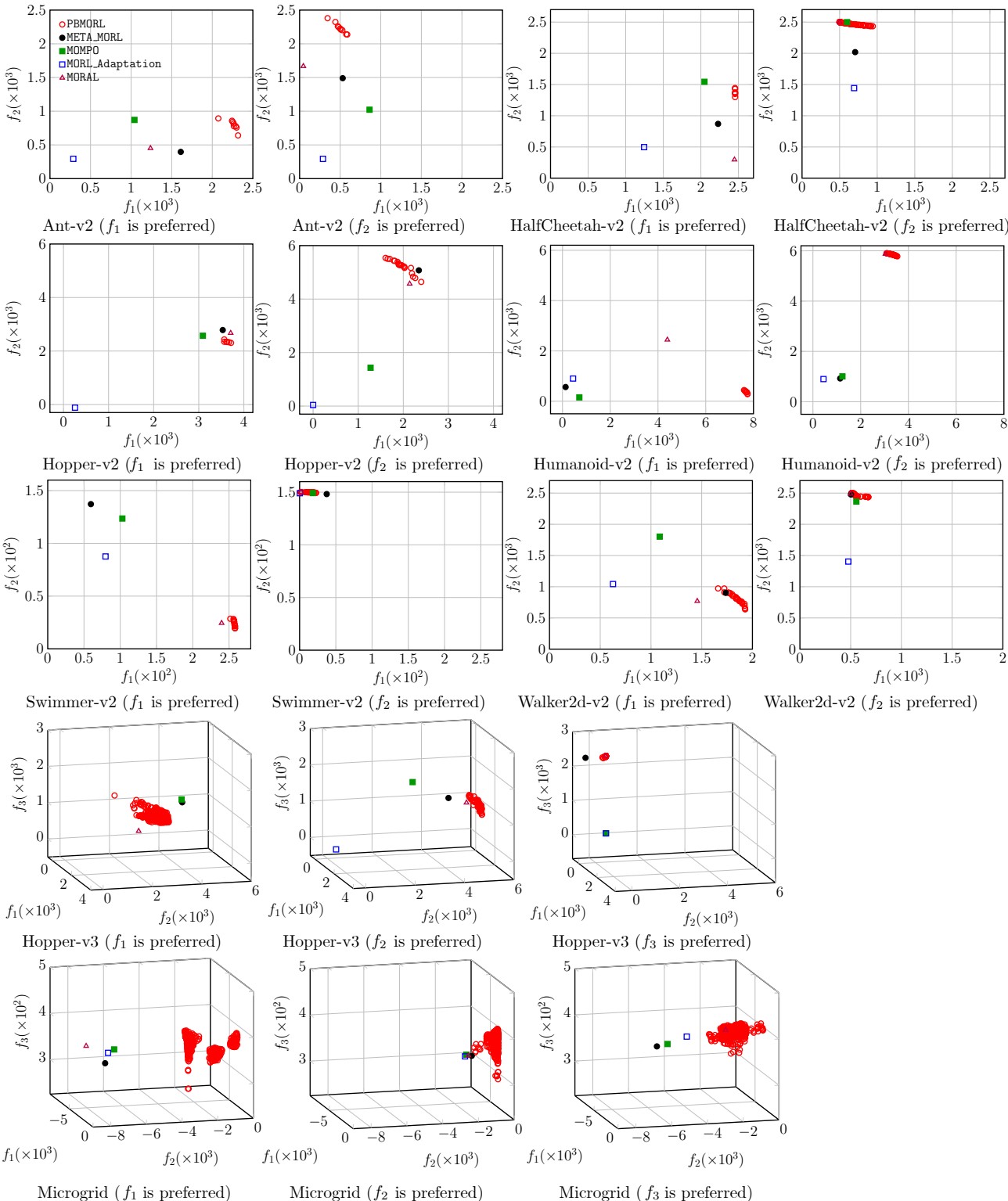

*Figure 10.* Plots of non-dominated policies obtained by `PBMORL` vs `MORL-Adaptation`, `META-MORL`, `MOMPO` and `MORAL`.

*Table 11.* Comparison results of $\epsilon^\star(\Pi)$ of PBMORL versus MOMPO, META-MORL, MORL-Adaptation and MORAL over 10 runs with mean and standard deviation.

| | | PBMORL | MOMPO | META-MORL | MORL-Adaptation | MORAL |
|---|---|---|---|---|---|---|
| Ant-v2 | $f_1$ | **7.709(1.68E−3)** | 8.910(1.81E−2) | 8.355(1.36E−2) | 9.686(8.53E−3) | 8.926(1.18E−2) |
| | $f_2$ | **7.637(6.98E−4)** | 8.914(2.05E−3) | 8.446(4.52E−3) | 9.633(3.37E−3) | 8.377(2.49E−3) |
| HalfCheetah-v2 | $f_1$ | **7.541(3.21E−6)** | 7.953(1.05E−4) | 7.752(1.77E−4) | 8.743(1.52E−4) | 7.545(9.73E−5) |
| | $f_2$ | **7.500(3.70E−9)** | **7.500(7.72E−8)** | 7.980(4.57E−8) | 8.561(1.55E−7) | **7.500(5.66E−8)** |
| Hopper-v2 | $f_1$ | **6.221(1.14E−2)** | 6.828(3.60E−3) | 6.404(3.54E−3) | 9.662(2.32E−3) | 6.236(2.42E−3) |
| | $f_2$ | **4.587(2.250E−2)** | 8.583(2.31E−2) | 4.930(4.68E−1) | 9.942(6.12E−2) | 9.761(7.94E−1) |
| Humanoid-v2 | $f_1$ | **2.362(2.51E−2)** | 9.292(4.17E−1) | 8.679(5.18E−2) | 9.554(8.33E−2) | 5.599(6.40E−2) |
| | $f_2$ | **4.099(6.96E−7)** | 8.993(1.84E−2) | 9.083(2.84E−3) | 9.912(8.34E−2) | 9.776(1.34E−2) |
| Swimmer-v2 | $f_1$ | **9.747(1.12E−4)** | 9.913(6.88E−4) | 9.933(5.58E−3) | 9.941(1.87E−3) | 9.759(2.99E−5) |
| | $f_2$ | **9.850(1.00E−12)** | **9.850(2.87E−9)** | 9.860(3.34E−7) | **9.850(1.09E−8)** | **9.850(7.22E−9)** |
| Walker2d-v2 | $f_1$ | **8.048(9.77E−4)** | 8.905(5.78E−3) | 8.267(1.26E−2) | 9.271(4.22E−3) | 8.850(6.31$E$-3) |
| | $f_2$ | **7.500(3.21E−8)** | 7.643(6.78E−3) | 7.528(2.39E−6) | 8.602(6.61E−3) | **7.500(1.92E−7)** |
| Hopper-v3 | $f_1$ | **6.020(8.91E−8)** | 6.821(2.91E−7) | 6.773(3.51E−5) | 9.070(8.77E−4) | 6.391(6.98E−6) |
| | $f_2$ | **4.346(2.02E−3)** | 7.295(1.72E−3) | 6.032(3.55E−3) | 9.411(3.87E−2) | 5.263(9.21E−3) |
| | $f_3$ | **7.500(1.90E−8)** | **7.500(2.63E−6)** | 7.522(3.96E−6) | **7.500(5.47E−7)** | **7.500(4.54E−7)** |
| MMSD | $f_1$ | **1.149(8.85E−4)** | 2.418(3.39E−3) | 2.737(9.05E−3) | 4.69(7.22E−3) | 2.48(6.72E−4) |
| | $f_2$ | **0.000(4.87E−7)** | 1.366(9.02E−4) | 1.317(8.62E−4) | 7.020(2.31E−4) | 0.672(5.56E−5) |
| | $f_3$ | **0.030(9.78E−8)** | 0.102(4.57E−6) | 0.071(8.76E−5) | 0.244(9.94E−4) | 0.124(1.29E−5) |

*Figure 11.* Plots of non-dominated policies obtained by PBMORL versus PGMORL, RA, and MOIA with different preferences.

## C.3. Comparison with conventional MORL

The goal of PBMORL is different from the conventional multi-policy MORL, which aims to approximate the entire Pareto front (PF). In contrast, we aim to identify the optimal policies align with DM's tacit preferences, which only represent a small portion of the PF. Particularly, such preferences can be some human intent, such as walking like a dog in our Unitree robot cases.

However, to more thoroughly evaluate our method, we compare our proposed PBMORL against four conventional MORL algorithms, which do not consider preferences, including PGMORL(Xu et al., 2020), PDMORL(Basaklar et al., 2023), RA (Parisi et al., 2014) and MOIA (Chiu et al., 2015). Note that since RA, PGMORL and PDMORL were merely designed for the MuJoCo environment while MOIA was deliberately designed for the MMSD environment, we only compare with them

*Table 12.* Comparison results of $\epsilon^{\star}(\Pi)$ and $\bar{\epsilon}(\Pi)$ of PBMORL versus PGMORL, RA and MOIA over 10 runs with mean and standard deviation.

| | | PBMORL | | PGMORL | | RA | | MOIA | |
|---|---|---|---|---|---|---|---|---|---|
| | | $\epsilon^{\star}(\Pi)$ | $\bar{\epsilon}(\Pi)$ | $\epsilon^{\star}(\Pi)$ | $\bar{\epsilon}(\Pi)$ | $\epsilon^{\star}(\Pi)$ | $\bar{\epsilon}(\Pi)$ | $\epsilon^{\star}(\Pi)$ | $\bar{\epsilon}(\Pi)$ |
| Ant-v2 | $f_1$ | **7.709(1.68E − 3)** | **7.880(3.32E − 2)** | 8.383(5.07E−3) | 8.898(7.39E−2) | 7.795(2.40E−3) | 8.506(6.13E−3) | | |
| | $f_2$ | **7.637(6.98E − 4)** | **7.751(1.65E − 4)** | 8.042(9.23E−3) | 8.322(1.57E−2) | 8.287(6.99E−3) | 8.965(1.66E−3) | | |
| HalfCheetah-v2 | $f_1$ | **7.541(3.21E − 6)** | **7.542(7.53E − 6)** | 7.554(1.84E−5) | 8.584(7.46E−4) | 7.542(1.14E−5) | 8.583(4.29E−4) | | |
| | $f_2$ | **7.500(3.70E − 9)** | **7.508(1.17E − 4)** | 7.506(2.31E−5) | 7.729(2.61E−4) | 7.504(2.44E−5) | 7.670(4.61E−5) | | |
| Hopper-v2 | $f_1$ | **6.221(1.14E − 2)** | **6.299(1.57E − 2)** | 6.744(9.76E−4) | 6.891(1.25E−1) | 6.467(3.78E−2) | 7.270(9.95E−2) | | |
| | $f_2$ | **4.587(2.25E − 2)** | **4.972(3.11E − 2)** | 6.160(8.05E−2) | 7.305(6.48E−2) | 4.903(3.99E−2) | 5.910(1.48E−3) | | |
| Humanoid-v2 | $f_1$ | **2.362(2.51E − 2)** | **2.583(1.49E − 1)** | 3.796(1.19E−2) | 5.066(2.95E−2) | 4.719(9.07E−2) | 5.783(1.80E−2) | | |
| | $f_2$ | **4.099(6.96E − 7)** | **4.143(9.79E − 5)** | 4.393(2.76E−2) | 5.06(6.93E−3) | 4.451(2.06E−2) | 4.67(1.54E−3) | | |
| Swimmer-v2 | $f_1$ | **9.747(1.12E − 4)** | **9.749(1.49E − 4)** | 9.753(1.25E−5) | 9.842(9.00E−5) | 9.756(4.32E−4) | 9.913(4.84E−4) | | |
| | $f_2$ | **9.850(1.00E − 12)** | **9.850(3.67E − 12)** | **9.850(1.08E − 7)** | 9.894(1.80E−6) | 9.852(1.11E−6) | 9.898(2.63E−5) | | |
| Walker2d-v2 | $f_1$ | **8.048(9.77E − 4)** | **8.116(3.95E − 3)** | 8.189(1.10E−2) | 8.870(2.90E−2) | 8.297(5.48E−3) | 9.110(1.23E−2) | | |
| | $f_2$ | **7.500(3.21E − 8)** | **7.506(8.84E − 5)** | **7.500(1.23E − 7)** | 7.786(3.84E−4) | 7.502(8.30E−6) | 7.835(2.54E−3) | | |
| Hopper-v3 | $f_1$ | **6.020(8.91E − 8)** | **6.203(1.44E − 3)** | 6.169(1.74E−3) | 7.56(4.22E−3) | 6.283(2.37E−3) | 8.331(1.55E−2) | | |
| | $f_2$ | **4.346(2.02E − 3)** | **4.539(1.97E − 2)** | 4.825(1.89E−3) | 7.131(3.34E−3) | 5.017(4.73E−3) | 8.428(2.40E−2) | | |
| | $f_3$ | **7.500(1.90E − 8)** | **7.502(1.23E − 5)** | **7.500(7.84E − 8)** | 8.201(4.15E−3) | 7.501(6.27E−7) | 7.984(3.80E−4) | | |
| MMSD | $f_1$ | **1.149(8.85E − 4)** | **1.846(3.88E − 5)** | | | | | 1.845(1.81E−3) | 3.902(2.56E−3) |
| | $f_2$ | **0.000(4.87E − 7)** | **0.511(3.41E − 6)** | | | | | 1.001(7.35E−7) | 5.051(4.09E−7) |
| | $f_3$ | **0.030(9.78E − 8)** | **0.050(5.74E − 8)** | | | | | 0.061(2.14E−7) | 0.090(5.22E−8) |

on their dedicated environment, respectively.

In addition to *approximation accuracy* $\epsilon^{\star}(\Pi)$, we introduce *average accuracy* $\bar{\epsilon}(\Pi)$ as an additional evaluation metric. The *average accuracy* measures the mean distance from all non-dominated policies in $\Pi$ to the DM-preferred reference policy in objective space, defined as:

$$\bar{\epsilon}(\Pi) = \frac{1}{|\Pi|} \sum_{\pi \in \Pi} \|\mathbf{J}(\pi) - \mathbf{J}(\pi^{\star})\|_2, \tag{13}$$

where $|\cdot|$ is the cardinality of a set.

We plot the non-dominated policies found by different algorithms in Fig. 11 and Fig. 12. Overall, the non-dominated policies obtained by RA and PGMORL approximate a wide range of the objective space, while the policies found by our proposed PBMORL are notably biased toward the DM's specified objectives, i.e., the ROI. This explains the comparable results of PBMORL and PGMORL on $\epsilon^{\star}(\Pi)$ in Table 12, where they even achieve the same mean $\epsilon^{\star}(\Pi)$ values in some cases. However, $\bar{\epsilon}(\Pi)$ shows that PBMORL provides better average alignment with the policies of interest, since RA and PGMORL identify many policies outside the ROI. For PDMORL, the visual comparison in Fig. 12 shows a similar pattern: PBMORL concentrates more strongly around the DM-preferred ROI across the MuJoCo tasks. For MMSD, the policies obtained by MOIA are dominated by those found by PBMORL, which is also reflected by the superior $\epsilon^{\star}(\Pi)$ and $\bar{\epsilon}(\Pi)$ values in Table 12.

### C.4. Golden policy

Note that calculating both $\epsilon^{\star}(\Pi)$ and $\bar{\epsilon}(\Pi)$ needs to specify a golden policy $\pi^{\star}$, which is represented as a $m$-dimensional hyperplane. Specifically, if we consider that the DM always prefers one objective over the others, $\pi^{\star}$ is defined as $f_i(\pi^{\star}) = 10,000$, where $i$ is the objective index preferred by the DM. As for the MMSD problem, we set $f_1(\pi^{\star}) = 0$ or $f_2(\pi^{\star}) = 0$ if the DM prefers the first or the second objective, while we set $f_3(\pi^{\star}) = 450$ if the DM prefers the third objective.

## D. Experimental Settings

This section introduces the settings of our empirical study, including the details about `complex reward` scheme, the benchmark problems, and the peer algorithms.

### D.1. The complex reward scheme for scalarized PPO

To better guide the robot's behavior, we introduce a series of auxiliary rewards for scalarized PPO, as detailed below.

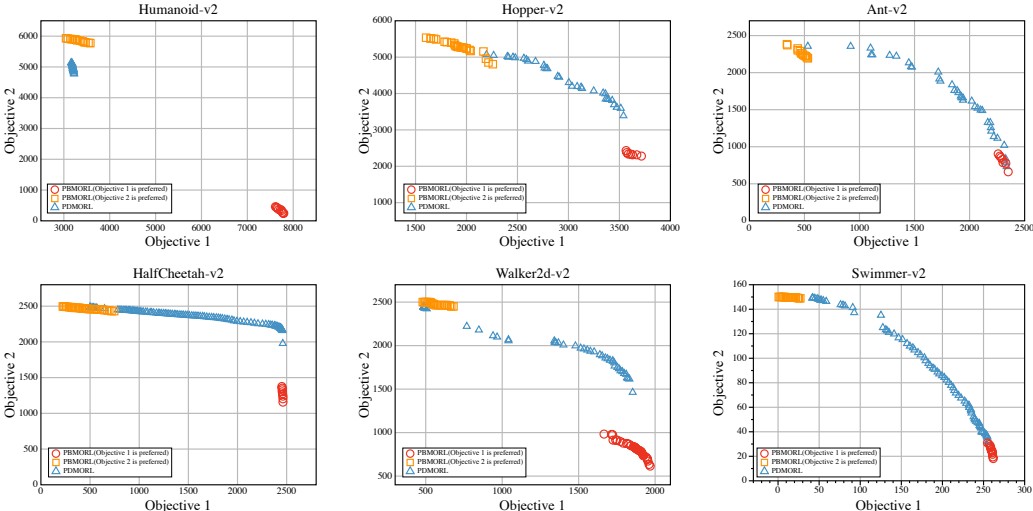

*Figure 12.* Plots of non-dominated policies obtained by `PBMORL` versus `PDMORL` on different MuJoCo tasks.

- **Linear Velocity in Z-axis** ($r_{\text{lin\_vel\_z}}$): Penalizes the robot's linear velocity along the vertical (Z) axis to encourage planar movement, ensuring stability and preventing unwanted vertical motion.

- **Angular Velocity in XY-axes** ($r_{\text{ang\_vel\_xy}}$): Penalizes the robot's angular velocity in the horizontal (X and Y) axes to promote smoother and more controlled rotations, enhancing overall balance.

- **Orientation** ($r_{\text{orientation}}$): Penalizes deviations from a flat base orientation by measuring the projection of gravity, thereby encouraging the robot to maintain an upright and stable posture.

- **Degrees of Freedom Acceleration** ($r_{\text{dof\_acc}}$): Penalizes the accelerations of the robot's joints to promote smooth transitions and reduce abrupt movements.

- **Action Rate** ($r_{\text{action\_rate}}$): Penalizes large changes in action commands between consecutive steps, fostering smoother and more consistent control actions.

- **Collision** ($r_{\text{collision}}$): Penalizes collisions on selected body parts by detecting significant contact forces, thereby encouraging the robot to navigate without unintended impacts.

- **Degrees of Freedom Position Limits** ($r_{\text{dof\_pos\_limits}}$): Penalizes joint positions that approach their mechanical limits, ensuring the robot operates within safe and feasible ranges.

- **Torque Limits** ($r_{\text{torque\_limits}}$): Penalizes torques that approach or exceed the robot's actuator capabilities, safeguarding the robot's mechanical integrity.

### D.2. Benchmark problems

In empirical study, we consider two types of benchmark problems: one is from the popular MUJOCO environment (Todorov et al., 2012) and the other is a multi-microgrid system design (MMSD) problem (Chiu et al., 2015).

#### D.2.1. MUJOCO ENVIRONMENT

Following (Xu et al., 2020), we examine seven benchmark problems developed from MUJOCO. Their objective functions and search spaces are outlined below:

- `Ant-v2`: This problem uses the speeds at the $x$ and $y$ axes as the two objectives. The state space is $\mathcal{S} \subseteq \mathbb{R}^{27}$, and the action space is $\mathcal{A} \subseteq \mathbb{R}^{8}$.

- `HalfCheetah-v2`: Two objectives are considered, including forward speed and energy consumption. The state space is $\mathcal{S} \subseteq \mathbb{R}^{17}$, and the action space is $\mathcal{A} \subseteq \mathbb{R}^{6}$.

- `Hopper-v2`: This problem considers forward speed and jumping height as the objectives. The state space is $\mathcal{S} \subseteq \mathbb{R}^{11}$, and the action space is $\mathcal{A} \subseteq \mathbb{R}^3$.

- `Humanoid-v2`: Two objectives are evaluated, including forward speed and energy consumption. The state space is $\mathcal{S} \subseteq \mathbb{R}^{376}$, and the action space is $\mathcal{A} \subseteq \mathbb{R}^{17}$.

- `Swimmer-v2`: This problem focuses on forward speed and energy consumption objectives. The state space is $\mathcal{S} \subseteq \mathbb{R}^8$, and the action space is $\mathcal{A} \subseteq \mathbb{R}^2$.

- `Walker2d-v2`: Two objectives are considered, including forward speed and energy consumption. The state space is $\mathcal{S} \subseteq \mathbb{R}^{17}$, and the action space is $\mathcal{A} \subseteq \mathbb{R}^6$.

- `Hopper-v3`: This problem incorporates three objectives, including forward speed, jumping height, and energy consumption. The state space is $\mathcal{S} \subseteq \mathbb{R}^{11}$, and the action space is $\mathcal{A} \subseteq \mathbb{R}^3$.

### D.2.2. MMSD ENVIRONMENT

Fig. 13 illustrates the microgrid environment, detailing the line flows and nodes within the power network. It considers the following three-objective optimization problem:

$$
\begin{aligned}
\text{maximize} \quad & \mathbf{r} = (\mathrm{U}_{\mathrm{pg}}(T), \mathrm{U}_{\mathrm{mg}}(T), \sum_{i=1}^{n_s} s_i(T))^\top \\
\text{subject to} \quad & \|s_i(t) - s_i(t-1)\| < c_i, \forall i \in \{1, \ldots, n_\mathrm{s}\}
\end{aligned}
\tag{14}
$$

where $\mathbf{r} \subseteq \mathbb{R}^3$ is the reward, $l_i < s_i(t) < u_i$. The mathematical definitions of these three objectives are delineated as follows, while the related notations are listed in Table 14.

- $\mathrm{U}_{\mathrm{pg}}(T)$ evaluates the utility value achieved by the power grid after one episode:

$$
\mathrm{U}_{\mathrm{pg}}(T) = \sum_{t=1}^{T} \sum_{i=1}^{n} \left( \mathrm{U}(p_i^d(t), w_i) - \lambda(t) p_i^d(t) \right),
\tag{15}
$$

  where

$$
\mathrm{U}(p_i^d(t), w_i) = 
\begin{cases}
\dfrac{w_i}{\alpha}, & \text{if } p_i^d(t) \geq \dfrac{w_i}{\alpha} \\
w_i p_i^d(t) - \dfrac{\alpha p_i^d(t)^2}{2}, & \text{if } 0 \leq p_i^d(t) \leq \dfrac{w_i}{\alpha}
\end{cases},
\tag{16}
$$

  where $w_i$ and $\alpha$ are pre-defined parameters.

- $\mathrm{U}_{\mathrm{mg}}(T)$ is the total utility value achieved by all microgrids after one episode:

$$
\mathrm{U}_{\mathrm{mg}}(T) = \sum_{t=1}^{T} \lambda(t) P_g(t) - \beta P_g(t)^2,
\tag{17}
$$

  where $\beta$ is a pre-defined parameter.

- To measure the stability of a multi-microgrid system, we use $\sum_{i=1}^{n_s} s_i(T)$ to evaluate its total energy storage. In particular, we have:

$$
s_i(t) = s_i(t-1) + p_i^g(t) - p_i^d(t) + v_i(t),
\tag{18}
$$

  where $p_i^d(t)$ is decided by both the base load of the $i$-th microgrid (Chiu et al., 2015) and a scaling factor:

$$
p_i^d(t) = (1 + h_i(t)) b_i(t),
\tag{19}
$$

  where $h_i(t)$ is defined as:

$$
h_i(t) = 
\begin{cases}
0.01\lambda^2(t) - 0.12\lambda(t) + 0.26, & \text{if } i = 1 \\
-0.01\lambda^2(t) + 0.13, & \text{if } i = 2 \\
-0.01\lambda^2(t) + 0.02\lambda(t) + 0.08, & \text{if } i = 3
\end{cases}.
\tag{20}
$$

The state space is $\mathcal{S} \subseteq \mathbb{R}^{n+2}$ where $\forall \mathbf{s} \in \mathcal{S}$, we have $\mathbf{s} = (t, p_1^g(t), \ldots, p_n^g(t), \lambda(t))^\top$. The action space is $\mathcal{A} \subseteq \mathbb{R}^{n_\mathrm{s}+1}$ where $\forall a \in \mathcal{A}$, we have $a = (\Delta\lambda(t), \Delta p_1^g(t), \ldots, \Delta p_{n_\mathrm{s}}^g(t))^\top$.

*Table 13.* List of the hyperparameter settings used in MuJoCo and MMSD.

| HYPERPARAMETER | MuJoCo ($m = 2$) | MuJoCo ($m = 3$) | Microgrid |
|---|---|---|---|
| # of environment steps | $8 \times 10^6$ | $8 \times 10^6$ | $4 \times 10^6$ |
| # of environment steps for the `seeding` stage | $4 \times 10^5$ | $4 \times 10^5$ | $4 \times 10^4$ |
| # of interactions with the DM | 40 | 40 | 40 |
| # of subtasks built before starting `PBMORL` | 6 | 21 | 21 |
| # of microgrids | | | 3 |
| # of microgrids with energy storage | | | 2 |

## D.3. Peer algorithms

In our experiments, we consider two types of peer algorithms. The first category consists of preference-based MORL algorithms from the literature:

- `MORL-Adaptation` (Yang et al., 2019): By generalizing the Bellman operator to MORL problems, `MORL-Adaptation` learns a universal parametric representation for all latent preferences. After the training phase, the learned policy can adapt to a given preference without further adaptation.

- `META-MORL` (Chen et al., 2019): `META-MORL` formulates MORL as a meta-learning problem conditioned on a task distribution over preferences. After the training phase, the learned policy can be adapted to a DM-specified preference through a fine-tuning phase.

- `MOMPO` (Abdolmaleki et al., 2020): `MOMPO` learns an action distribution for each objective in each round of policy training and updates the policy by fitting it to a combination of action distributions. Before the training phase, a set of parameters representing the DM's preference for each objective is provided, controlling the learning rate of the corresponding action distribution.

- `MORAL` (Peschl et al., 2022): `MORAL` first learns a multi-objective reward function from demonstrations. Then, similar to our proposed `PBMORL`, it learns the DM's preference as a weight vector based on the Bradley-Terry model (Bradley & Terry, 1952) to guide the policy optimization.

The second category comprises conventional MORL algorithms that do not take the DM's preference information into account.

- `RA` (Parisi et al., 2014): `RA` transforms a MORL problem into several single-objective RL tasks by weighted aggregations. Different from our proposed `PBMORL`, `RA` solves these tasks independently.

- `MOIA` (Chiu et al., 2015): `MOIA` is a dedicated multi-objective evolutionary algorithm tailored for the multi-microgrid system design problem.

- `PGMORL` (Xu et al., 2020): Designed to optimize multiple policies in parallel, `PGMORL` iteratively adjusts the directions based on a predictive method.

All peer algorithms used the same number of environment steps, and Table 13 lists the key hyperparameters used in our experiments.

## E. Discussion

### E.1. Time complexity analysis

Since the `seeding` and the `policy optimization` modules of `PBMORL` are MORL based on PPO, the corresponding time complexity is bounded by the PPO itself. Therefore, here we focus on analyzing the time complexity of the `preference elicitation` module, which consists of three steps. Specifically, during the `consultation` step, the preference model first evaluate the quality of policies in $\Pi$ which incurs $\mathcal{O}(|\Pi|)$ evaluations. Then, it chooses two elite policies to query the DM. After the DM's feedback is collected, the complexity of the `preference learning` step is

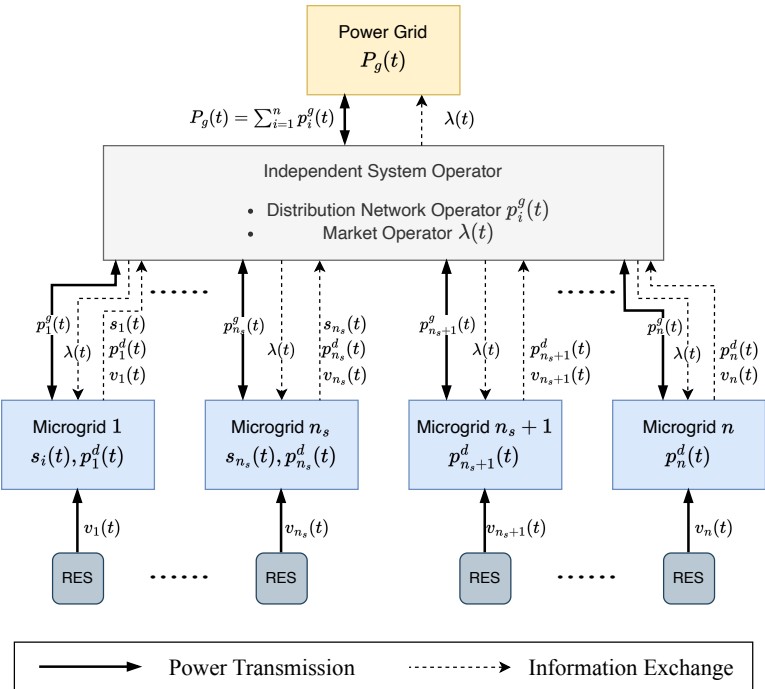

*Figure 13.* System operation model of the network of microgrids.

*Table 14.* Lookup table of the mathematical notations used in the MMSD problem (Chiu et al., 2015).

| SYMBOL | DESCRIPTION |
| --- | --- |
| $n$ | The number of microgrids |
| $n_{\mathrm{s}}$ | The number of microgrids with power storage |
| $t$ | The current time step |
| $T$ | The length of one episode |
| $P_g(t)$ | The sum of power given to all of microgrids at the $t$-th time step |
| $p_i^g(t)$ | The power given to the $i$-th microgrid at $t$-th time step |
| $p_i^d(t)$ | The power demand of the $i$-th microgrid at the $t$-th time step |
| $\lambda(t)$ | The power price at the $t$-th time step |
| $s_i(t)$ | The power storage of the $i$-th microgrid at the $t$-th time step |
| $v_i(t)$ | The energy that the distributed power gives to the $i$-th microgrid at the $t$-th time step |
| $b_i(t)$ | The base load of the $i$-th microgrid at the $t$-th time step |
| $c_i$ | The maximum rate of storage charging and discharging |
| $l_i$ | The lower bound of the energy storage of the $i$-th microgrid |
| $u_i$ | The upper bound of the energy storage of the $i$-th microgrid |

bounded by $\mathcal{O}(\kappa^3)$, where $\kappa$ is the number of data instances in the training set. During the `preference translation` step, the computational complexity is mainly dominated by the ranking of policies in $\Pi$, i.e., $\mathcal{O}(|\Pi| \log |\Pi|)$. All in all, the time complexity of the `preference elicitation` module is $\max \left\{ \mathcal{O}(\kappa^3), \mathcal{O}(|\Pi| \log |\Pi|) \right\}$.

### E.2. MORL formulation

Our formulation follows the scalarized expected return (SER) perspective: the preference model is learned over objective-value vectors $\mathbf{J}(\pi)$, and policy improvement optimizes scalarized expected returns for decomposed subproblems. The expected scalarized return (ESR) formulation would instead apply a nonlinear utility at the trajectory-return level before expectation. ESR generally breaks the additive Bellman structure exploited by standard PPO-style policy optimization (Hayes et al., 2022). Extending PBMORL to directly optimize ESR objectives is therefore outside the scope of this work and is an interesting direction for future research.

### E.3. Future directions

Human-in-the-loop interactive MORL presents a promising paradigm for realizing human-AI collaboration, but several limitations remain. First, the current framework assumes that the DM's preference is stationary during one optimization run. Appendix B.5 studies fuzzy or noisy preferences, but it does not address genuinely time-varying preferences that evolve during deployment. Detecting and adapting to such preference drift is an important future direction. Second, this paper assumes that the information provided by MORL is fully understandable by the DM, which may not be realistic, particularly when dealing with more than three objective functions (similar discussion can be found in (Li et al., 2018a)). Developing a human-computer interaction platform and mechanism is essential for enhancing the effectiveness of interactive MORL. It is also interesting to explore the application of RL in the context of evolutionary multi-objective optimization (Zhang et al., 2024). Finally, the explainability of the policies of interest and their implications has rarely been discussed in the literature, representing another area for future investigation.

