# OpenReview forum: "Human-in-the-Loop Policy Optimization for Preference-Based Multi-Objective Reinforcement Learning"
_ICML.cc/2026/Conference — ICML 2026 regular_

### Official Review · Reviewer_HTXx · 2026-03-02

**Soundness:** 3
**Presentation:** 3
**Significance:** 3
**Originality:** 3
**Overall Recommendation:** 5
**Confidence:** 4

**Summary:**

This work introduces a human-in-the-loop MORL framework. It addresses the challenges of overwhelming decision-makers with too many policies and the difficulty of specifying preferences in advance. The framework interactively learns the Decision Maker's implicit preferences during optimization, without requiring prior knowledge, and integrates this learning into a parallel search process to find high-quality, preference-aligned policies. Experiments in robotics simulation and other domains demonstrate its effectiveness compared to state-of-the-art methods.

**Compliance With Llm Reviewing Policy:**

Affirmed.

**Final Justification:**

The questions of my review have been answered.

**Key Questions For Authors:**

In tasks with severe objective conflicts or highly sparse environments, can the brief, preference-agnostic parallel exploration (seeding phase) effectively generate "seed" policies that cover key regions of the Pareto front? If the seed quality is poor, could it "contaminate" subsequent preference learning and cause the search to get trapped in a local optimum?
The linear weight-biasing mechanism in Eq. 7 may trap the search around initial, possibly inaccurate preference anchors, causing convergence to a suboptimal "preference local optimum" instead of the true region of interest？

**Limitations:**

yes

**Strengths And Weaknesses:**

Strengths
It overcomes two major limitations of traditional MORL methods—namely, the impracticality of requiring preferences to be specified in advance and the burden of generating numerous irrelevant policies—through interactive, human-in-the-loop learning.
It innovatively and tightly couples preference learning with the multi-policy optimization process. It dynamically guides parallel task allocation using preferences learned in real time, enabling the efficient discovery of high-quality policies aligned with the decision maker's preferences with only a small number of interactions.
Weaknesses
The core framework assumes the decision maker's preferences are static and deterministic. This assumption limits its applicability to more complex, real-world scenarios involving dynamic or ambiguous preferences.

---

> ### Author Rebuttal · Authors · 2026-03-30
>
> We sincerely thank the reviewer for their supportive assessment and thoughtful questions. We summarize the reviewer’s concerns and address each of them in detail below.
>
> **W1. Dynamic or ambiguous preferences.**
>
> This is an excellent point regarding a critical challenge in real-world systems. We address it from two aspects:
>
> 1. **Ambiguous preferences**. We explicitly study robustness to non-deterministic preferences in our *fuzzy preference* setting (**Appendix B.5**). The results show that even when the DM's preference is stochastic, our method successfully converges to a policy that reflects the DM's dominant preference trend, rather than being misled by noisy individual queries.
> 2. **Time-varying preferences.** We agree this is an important limitation. The current framework assumes *stationary* preferences. While Appendix B.5 addresses ambiguity/noise, it does not cover genuinely dynamic preferences that evolve over time. Extending PBMORL to handle non-stationary, time-varying preferences is a promising direction for future work. We will add this discussion in the camera-ready version.
>
> **Q1. Impact of seed quality**
>
> This is an excellent and critical question. We respond from three perspectives:
>
> 1. First, unlike the reviewer’s assumption, we do not require the seeding policies to cover key regions of the Pareto front. In most cases, these policies are still far from the Pareto front, but are better than random policies. The purpose of the seeding phase is to help policies quickly move away from noisy random behaviors. This provides more informative comparisons for the DM and a better starting point for subsequent optimization.
> 2. We have included an ablation study in **Appendix B.8** (Fig. 9). The results show that the impact of seeding depends on task complexity. In simpler environments (e.g., Swimmer-v2), the seeding stage has minimal effect, whereas in more complex tasks (e.g., Humanoid-v2), better seeds lead to faster convergence. Importantly, PBMORL remains robust with a short seeding phase (5% of total budget), even on challenging tasks such as Humanoid. We acknowledge that for highly complex real-world tasks, improving seed quality (e.g., by increasing the seeding budget or using stronger RL optimizers) can further accelerate convergence.
> 3. PBMORL does not rely on the initial seeds as fixed anchors. The candidate policy set is continuously updated using non-dominated filtering, allowing newly discovered policies to replace suboptimal ones. Moreover, preference guidance is **not tied to a single weight vector**, but translated into a **distribution over tasks** (Sec. 3.2.3). The algorithm explicitly combines exploitation and exploration tasks, ensuring that the search continues to explore nearby trade-offs and can correct early inaccuracies in seed policies or preference estimates, rather than being trapped around initial anchors.
>
> Consistent with this concern, our own ablations show that retaining nonzero exploration is important ($\kappa_2>0$ in Appendix B.4), and overly aggressive biasing ($\eta=1.0$) degrades performance substantially (Table 9), which is precisely why PBMORL mixes exploitation and exploration rather than collapsing around a single anchor.

---

> > ### Author Rebuttal · Reviewer_HTXx · 2026-04-01
> >
> > I have reviewed the authors' rebuttal and find that it addresses my key concerns regarding seed quality robustness and preference ambiguity. The explanations and references to provided ablation studies are satisfactory. Therefore, I maintain my original recommendation.

---

> > > ### Author Response · Authors · 2026-04-05
> > >
> > > Thank you again for the careful reading and constructive discussion. We appreciate the feedback and will incorporate these clarifications in the final revision.

---

### Official Review · Reviewer_Ti1A · 2026-03-06

**Soundness:** 3
**Presentation:** 3
**Significance:** 3
**Originality:** 3
**Overall Recommendation:** 5
**Confidence:** 2

**Summary:**

This paper proposes a novel human-in-the-loop method for multi-objective reinforcement learning. The core idea is both novel and interesting, effectively addressing the challenge of non-linear combinations of multiple reward functions, a limitation faced by a number of prior multi-objective methods. The theoretical derivation is sound, and the experimental evaluation is thorough and solid.

**Compliance With Llm Reviewing Policy:**

Affirmed.

**Key Questions For Authors:**

Since the method involves a human-in-the-loop framework, could you please clarify how users are expected to participate in practice? For instance, it would be helpful to specify the frequency with which humans should annotate the trained policy and the amount of time or effort required from them. Such details are essential for assessing the practical applicability of the proposed method.

**Limitations:**

Please refer to the weakness and questions.

**Strengths And Weaknesses:**

Strngths:
1. The proposed method is novel and interesting, representing the first attempt to introduce human input into the multi-objective reinforcement learning scenario. It effectively addresses the difficulty of designing coefficients for multiple objectives by accounting for the black-box nature of human preferences.
2. The authors provide a detailed and in-depth analysis of the method through experimental results, offering sufficient insights to guide future research in this direction.

Weakness:
1. Since the paper focuses on human preference, a user study, while not strictly necessary, would have further strengthened the validation of the proposed approach.

---

> ### Author Rebuttal · Authors · 2026-03-30
>
> We sincerely thank the reviewer for their thoughtful comments. We are encouraged by their positive assessment of our methodology, technical soundness and experimental evaluation. Below, we summarize the reviewer’s concerns and provide our responses.
>
> **W1. Human feedback**
>
> We agree that a broader user study would further strengthen the paper. Our current real-human Unitree case study (**Section 4.1**) is a first step in that direction.
>
> **Q1. Interaction frequency**
>
> In PBMORL, human involvement is designed to be lightweight and practical.
>
> - As discussed in **Section 3.2.1**, in each consultation round, the user performs **one pairwise comparison** between two short policy rollouts (better / worse / indifferent), which is lightweight and intuitive.
> - In our main experiments, we use **40 queries**, while **Appendix B.1** shows that **10–20 queries** are often sufficient to achieve strong performance.
>
> In our Unitree case study, each comparison involved choosing between two short rollouts and typically took only a few seconds; thus 40 queries required only a few minutes of human effort.

---

> > ### Author Rebuttal · Reviewer_Ti1A · 2026-04-02
> >
> > Thank the authors for the response. I believe this is a good and practical method. Therefore, I maintain my score.

---

> > > ### Author Response · Authors · 2026-04-05
> > >
> > > Thank you again for the careful reading and constructive discussion. We appreciate the feedback and will incorporate these clarifications in the final revision.

---

### Official Review · Reviewer_xdxk · 2026-03-11

**Soundness:** 3
**Presentation:** 4
**Significance:** 3
**Originality:** 3
**Overall Recommendation:** 4
**Confidence:** 3

**Summary:**

This paper investigated the massive inefficiency in standard Multi-Objective Reinforcement Learning pipelines, where real human preferences remained implicit and resisted explicit numerical quantification. The authors propose the PBMORL framework to dynamically couple active preference learning with policy optimization and experiments on continuous control and microgrid environments validate the performances.

**Compliance With Llm Reviewing Policy:**

Affirmed.

**Final Justification:**

My concerns are correctly addressed.

**Key Questions For Authors:**

See weaknesses

**Limitations:**

yes

**Strengths And Weaknesses:**

Strengths:
- The writing is clear and easy to understand
- The method completely avoided a naive two-stage apporach, the deep coupling of preference elicitation and policy optimization looks reasonable and works great.
- Experimental results are convincing.

Weaknesses:
- (Major) The human-in-the-loop experiments used simulated human feedbacks which fundamentally weakens the claims
- (Major) Though outperform baselines, the efficiency should also be considered, as complexity analysis is offered in Appendix, whether experimental results support the claim is unknown.
- (Minor) The problem itself is not novel, preference-based RL/online learning/multi-objective online learning has been studied with varies kind of theoretical guarantees, while current technical tools look standard (PO and GP), there's no obvious intuition or theoretical guarantees explaining why they work so well.
- (Minor) The components of the algorithm is claimed to be flexible and easy to replaced, but I'm not sure if I'm getting it correctly, the preference elicitation is not decoupled with certain policy optimization methods.

---

> ### Author Rebuttal · Authors · 2026-03-30
>
> We sincerely thank the reviewers for their detailed and valuable comments, as well as for their positive assessment of our work. We summarize the reviewer’s concerns below and provide our responses accordingly.
>
> **W1. Human feedback**
>
> We agree the manuscript did not clearly separate the two evaluation regimes.
>
> - Unitree is a real-human case study with pairwise human comparisons (**Sec 4.1**, lines 235-245) rather than simulated feedback.
> - For the MuJoCo and MMSD experiments, we use simulated preference oracles. This follows standard practice in preference-based RL (Christiano et al., 2017), where algorithmic oracles serve as reproducible proxies for human preferences. Importantly, running the full benchmark suite across baselines and seeds already requires hundreds of training runs, and substantially more when including ablations. These make real human evaluation logistically infeasible at scale.
>
> We will revise the abstract and Sec. 4 to explicitly distinguish the real-human Unitree case study from the oracle-based large-scale benchmarking suite, and narrow the wording accordingly.
>
> **W2. Efficiency**
>
> We thank the reviewer for raising this important point. We address efficiency from three perspectives:
>
> 1. **Human-query efficiency:** PBMORL requires only a small number of human interactions. In the main experiments, only **40 queries** are used, and **Appendix B.1** shows that strong performance is maintained with **10–20 queries**.
> 2. **Environment-sample fairness:** All peer algorithms in Sec. 4.2 are compared under the same environment-interaction budget, ensuring fair comparison in RL computation cost (**Appendix D.3**).
> 3. **Extra compute overhead:** The extra cost of PBMORL comes from preference elicitation, whose complexity is analyzed in **Appendix E.1** as $\max\\{\mathcal{O}(\kappa^3),\mathcal{O}(|\Pi|\log|\Pi|)\\}$. In practice, $\kappa$ is very small (only $40$ queries), making this overhead negligible compared to PPO rollouts and updates. In our implementation, **a single GP update takes less than $1$ second, while one round of PPO training requires ~$15$ minutes**. This further confirms that the additional computation from preference learning is negligible in practice.
>
> **W3. Novelty**
>
> We agree that GP and PPO are standard components, and our contribution is algorithmic rather than theoretical. The novelty lies in their closed-loop coupling: (i) uncertainty-aware pairwise preference learning over candidate policies, and (ii) dynamic reallocation of the multi-policy optimization budget toward a learned region of interest (ROI). The ablations in **Appendices B.2**, **B.4**, and **B.7** further support that both uncertainty estimation and the ROI-biased task distribution are empirically important.
>
> **W4. Modularity of PBMORL**
>
> PBMORL is modular at the interface level: the preference learner is required to provide utility estimates (ideally with uncertainty), while the optimizer must support solving a collection of scalarized or decomposed subproblems. Within this interface, the framework is flexible across both preference models and aggregation/optimization choices.
>
> In **Appendix B.2**, we replace the GP with alternative preference learning methods, including ranking-SVM (Joachims, 2002) and a preference learning method in RL (Wirth et al., 2016). In **Appendix B.6**, we replace the linear aggregation with Tchebycheff aggregation. These variants consistently achieve competitive performance, demonstrating that PBMORL is not tied to specific implementations of either module.

---

> > ### Author Rebuttal · Reviewer_xdxk · 2026-04-05
> >
> > Thank you for the clarification. My concerns are correctly addressed so I'm willing to raise the score.

---

> > > ### Author Response · Authors · 2026-04-05
> > >
> > > Thank you again for the careful reading and constructive discussion. We appreciate the feedback and will incorporate these clarifications in the final revision.

---

### Official Review · Reviewer_K3tQ · 2026-03-12

**Soundness:** 2
**Presentation:** 1
**Significance:** 2
**Originality:** 3
**Overall Recommendation:** 4
**Confidence:** 4

**Summary:**

This paper proposes a multi-objective reinforcement learning (MORL) method with a human-in-the-loop called PBMORL. The approach maintains a candidate pool of policies and generates linear weight vectors that represent preferences over the objectives in order to guide exploration of the Pareto front using pairwise feedback from a human decision-maker (DM). Given the DM’s preferences between policies in the pool, the method models the user’s utility function with a Gaussian Process posterior. This learned utility model is then used to generate a set of weight vectors that guide the next iteration of policy optimization. The goal of the method is to iteratively learn the DM’s implicit preferences while optimizing policies, without requiring the DM to specify reward weights beforehand. The framework alternates between preference elicitation and policy optimization, progressively focusing the search on regions of the Pareto front that better align with the DM’s preferences. Experiments are conducted on high-dimensional continuous control tasks and compared against several MORL baselines, including both preference-based approaches and conventional methods that approximate a Pareto frontier.

**Compliance With Llm Reviewing Policy:**

Affirmed.

**Final Justification:**

I have increased my score to 4 after the author's clarifications during the rebuttal. I am not confident I can further raise my score due to the presentation issues raised in my review and the paper's clarity issues regarding the experimental setup.

**Key Questions For Authors:**

- Although I assume that the method’s acronym means Preference-Based MORL, this is never explicitly defined in the paper.


- Line 72 (right column): $T$ should be defined as a function. As written, it appears to be a scalar value.


- The nomenclature “task” for different weight vectors in MORL is unusual. In general, there is a single task, and the agent solves it using different preferences over the objectives.


- In Section 3.2.2, the authors assume the scalarized expected return (SER) formulation for non-linear utilities (Hayes et al., 2022). It would also be useful to discuss the expected scalarized return (ESR) formulation.


- In Eq. (3), $\Phi$ and $o_i$​ are not defined. Some intuition for this equation would also help the reader.


- In Eq. (4) and throughout the rest of the section, several variables are not clearly defined. This section would benefit from clearer notation and explanation.


- Section 4.1: It is unclear how the human feedback was introduced. Was there an actual user providing preferences, or were preferences simulated with a program?


- The method introduced in Yang et al. (2019) is called Envelope Q-learning, not “MORL-Adaptation”.


- In Section 4.2, it is not clear what the DM preferences are and how they are provided to each method. PBMORL uses pairwise feedback between policies. Do the other algorithms receive the same type and amount of feedback?

**Limitations:**

One limitation that should be discussed is the computational cost associated with maintaining and optimizing multiple policies in parallel. Although the approach focuses exploration on preference-relevant regions, the framework still requires training several policies simultaneously using RL, which may be expensive in large-scale environments.

**Strengths And Weaknesses:**

### Strengths

- The paper tackles an important problem in MORL: incorporating human preferences during training rather than requiring them to be specified in advance.


- The approach proposed that uses pairwise preference feedback and a Gaussian Process model to learn a utility function from limited human input seems to be novel and sound.


- The experiments evaluated the approach on several continuous control benchmarks and a complex robotics environment.

### Weaknesses

- The authors claim that they do not assume that the decision-maker’s (DM) preference can be represented by a true scalarization weight vector and state that human preferences cannot be captured by a linear weighting. However, the algorithm still relies on linear scalarization to parameterize the search, which may limit its ability to represent arbitrary nonlinear preferences. If that is correct, the beginning of Section 3 would be somewhat misleading, since it states that the method is not looking for a “true” scalarization vector.


Some claims in the experiments are not fully supported by the presented results:
- First, the authors claim that PBMORL policies better match intended behaviors compared to scalarized PPO. However, the comparison does not appear to be fair, since PPO was trained with only five weight vectors, whereas PBMORL can train with many weight vectors generated by Eq. (7). A fairer baseline would allow PPO to train with the same number of weight vectors as PBMORL, sampled randomly.


- The authors state that PBMORL found policies that dominate scalarized PPO policies. It is unclear whether both methods used the same training budget. Since PBMORL itself uses PPO to optimize policies, this result is not intuitive. If both methods use the same PPO implementation and training budget, policies optimized with PPO for a given weight vector should not be dominated by each other. In other words, since PBMORL internally uses PPO, it is unclear why scalarized PPO policies are dominated unless the training budgets or exploration strategies differ.


- Regarding point 5 in Section 4.1.2, the authors show that a policy with a higher weight for energy efficiency was actually less energy-efficient than another policy with a smaller weight in a problem with two objectives. This likely indicates that training had not converged or that the results are affected by stochasticity. Assuming a perfect policy optimization algorithm, this result should be impossible. These results should be reported with mean values and confidence intervals, and the number of random seeds used should be stated.

---

> ### Author Rebuttal · Authors · 2026-03-30
>
> We thank the reviewer for the valuable comments and for recognizing our novelty and soundness.
>
> **W1. Scalarization as search coordinate parameterization**
>
> 1. Our main claim is that PBMORL does not assume DM can provide a single “**true**” weight vector a priori. PBMORL learns a **nonlinear utility over objective vectors $J(\pi)$, while using scalarization only as a search parameterization**.
> 2. PBMORL is not conceptually restricted to a linear weighted-sum aggregation; **App. B.6** shows that a Tchebycheff variant can also be used.
> 3. We agree that the current implementation is still constrained by the chosen objective space and its weight-based search parameterization. We will narrow the wording in Sec 3 accordingly.
>
> **W2. PPO with broader weight coverage**
>
> To address this concern, we ran a stronger dense fixed-weight PPO baseline, with **21 uniformly sampled static weights** (covering the 2-objective simplex at 0.05 resolution) under the same per-run budget. Each weight was trained with 2 seeds, resulting in a total budget **42$\times$ larger** than a single PBMORL run. Results are shown in **Fig. R1** (see https://tinyurl.com/2s4229ay).
>
> Despite dense weight coverage, PPO still fails to achieve good trade-offs across objectives. This suggests that broader weight coverage alone is insufficient, and that PBMORL’s preference-guided exploration/task reallocation is an important source of the gain.
>
> **W3. Training budget and why PBMORL outperforms PPO**
>
> In Sec 4.1, each PBMORL run and each PPO run used the same per-run environment budget of $10^7$ steps. The advantage of PBMORL stems from its effective integration of preference learning and guided exploration, rather than increased budget or more weight vectors.
>
> For why PBMORL outperforms PPO:
>
> 1. Deep RL policy optimization is inherently non-convex, and PPO is **not** a perfect optimizer. Linearly combining conflicting rewards radically alters the loss landscape, often trapping scalarized PPO in poor local optima (e.g., the robot falls over instantly to minimize energy).
> 2. For the same reason, the relationship between reward weights and resulting policy performance is highly nonlinear (Hayes et al., 2022). As in Fig. 2A, favorable trade-offs are not reliably recovered via weight tuning.
> 3. PBMORL uses preference-guided adaptive exploration (Sec 3.2.3) to steer search toward a region of interest while maintaining diversity, enabling escape from local optima that trap isolated PPO runs.
>
> The PPO implementation is the same base optimizer; the gain comes from warm-started multi-policy search and adaptive task reallocation, not from a stronger inner optimizer.
>
> **W4. Statistics**
>
> We agree that the original point 5 was too categorical. The new seed statistics suggest that this particular pair should be viewed as an illustration of practical non-monotonicity rather than a statistically decisive statement. We will revise the text and report mean $\pm$ std/95% CI over 5 independent training seeds in **Table R1** (see https://sites.google.com/view/pbmorl-re).
>
> **Q1**: PBMORL = Preference-Based MORL.
>
> **Q2**: $T$ is the transition function $T(s’|s,a)$. We will define it explicitly.
>
> **Q3**: We agree “task” is nonstandard here; we mean scalarized subproblem and will rename it.
>
> **Q4**: Our current framework targets SER; ESR under nonlinear utility generally breaks the standard Bellman-additive structure assumed by PPO optimization. We will add the discussion to Sec 3.
>
> **Q5**: $\Phi$ is the standard normal CDF; $o_i$ is the $i$-th pairwise comparison label. Eq. (3) is the probit likelihood of a pairwise comparison: the probability of $o_i$ increases as the latent utility gap between the two compared policies increases.
>
> **Q6:** In Eqs. (3)-(6), $\tilde{\Pi}$ denotes the set of unique policies appearing in the preference dataset $D$, and $K_{\tilde{\Pi}}$ is the kernel matrix over $\tilde{\Pi}$. $\mathbf{U}=\mathbf{u}^\star$ denotes the MAP estimate of latent utilities over $\tilde{\Pi}$. $\mathbf{k}_{\pi}$ is the covariance between $\pi$ and policies in $\tilde{\Pi}$, and $Λ$ is the MAP estimate's uncertainty.
>
> **Q7**: Sec 4.1 (Unitree) uses real human pairwise feedback under three instructed preference profiles; Sec 4.2 (MuJoCo/MMSD) uses oracle/simulated preferences for controlled benchmarking.
>
> **Q8**: Thank you; we will correct this to Envelope Q-learning throughout.
>
> **Q9**: PBMORL is the only method in our comparison that uses online pairwise policy feedback during training. The other baselines follow their native preference interfaces instead of receiving the same type/amount of feedback (**App. D.3**).
>
> **Limitations**: We agree that maintaining multiple policies increases RL-side compute. Our point is not that PBMORL is cheaper than single-policy PPO, but that its additional preference-learning overhead is negligible relative to PPO, and its ROI-focused search avoids wasting compute on irrelevant parts of the Pareto front.

---

> > ### Author Rebuttal · Reviewer_K3tQ · 2026-04-02
> >
> > ### W1
> > "PBMORL learns a nonlinear utility over objective vectors $J(\pi)$, while using scalarization only as a search parameterization."
> > I am still not convinced by this claim. Although PBMORL might learn a non-linear utility, no policy in its pool actually optimizes this utility. All policies learned by PBMORL are policies specialized to a linear weight scalarization, and are not specialized to the learned non-linear utility.
> >
> > ### W2:
> > "Each weight was trained with 2 seeds, resulting in a total budget 42$\times$ larger than a single PBMORL run."
> > The number of random seeds has nothing to do with the computation budget. Seeds are used to replicate an experiment N times and report the mean performance with confidence levels.
> > The authors trained 21 individual PPO agents (one agent per weight) and also executed a PBMORL run. It is still unclear:
> > - What is the training budget per run? How many weights does PBMORL train in total? For instance, does PBMORL also use 21 different weights? How is the total training budget divided per weight? The total training budget should be the same for PPO and PBMORL.

---

> > > ### Author Response · Authors · 2026-04-02
> > >
> > > Thank you. These are fair follow-up questions, and we agree our previous wording was not precise enough.
> > >
> > > **Reply to W1.** You are right to distinguish **direct optimization of the learned nonlinear utility** from **utility-guided search**. In the current PBMORL, the learned nonlinear utility over objective vectors $J(\pi)$ is **not** used as the per-policy RL objective. Individual policy updates are still performed on scalarized subproblems. The role of the learned utility is at the **outer loop**: it scores candidate outcomes, defines the ROI, and is translated into an adaptive distribution over scalarized subproblems that receive further optimization/comparison. Thus, PBMORL is more accurately characterized as a **utility-guided, decomposition-based search method**, rather than a direct optimizer of an arbitrary nonlinear utility. We will revise the wording in Sec. 3 accordingly.
> > >
> > > **Reply to W2.** You are also right that our previous wording conflated **seed replication** with **optimization budget**. To clarify:
> > >
> > > 1. One fixed-weight PPO run uses $10^7$ environment steps.
> > > 2. The dense PPO sweep over **21 fixed weights** therefore uses **$21 \times 10^7$** environment steps in aggregate; the second seed was used **only for variance estimation**, not as part of the method budget.
> > > 3. One PBMORL run also uses $10^7$ environment steps **in total across the entire adaptive run**, not $10^7$ per translated weight.
> > >
> > > In PBMORL, translated weights are **not** trained as independent full-budget PPO jobs. More specifically, the seeding phase uses **6 initial weights/policies** with **5%** of the total budget. Afterwards, as in **Sec. 3.2.3**, each round maintains a fixed-size active set of subproblems: $\kappa_1=2$ exploitation subproblems and $\kappa_2=8$ exploration subproblems. The **10 reference weights** $\tilde{W}$ are only used to define and bias the sampling distribution around the ROI; they are **not** 10 fresh full-budget PPO runs. Exploration subproblems are warm-started from nearby promising policies in $\dot{\Pi}$, and the single global **$10^7$**-step budget is reallocated across these active subproblems over time. Thus, “budget per weight” is not analogous to 21 independent fixed-weight PPO agents.
> > >
> > > With this distinction, the original Fig. 2/Table 1 comparison is a **matched per-run-budget** comparison, whereas the 21-weight PPO sweep is a **stronger aggregate-compute coverage baseline** that we added specifically to address the concern that five static weights might be too sparse.

---

### Decision · Program_Chairs · 2026-04-30

**Decision:**

Accept (regular)

**Comment:**

This work integrates human preference learning into the multi-objective optimization process. Reviewers acknowledge its significance and soundness. The in-depth analysis of the method, supported by comprehensive experimental results, is convincing. During the rebuttal period, the authors addressed concerns regarding the seeding policy and the stochasticity of preferences, and clarified a practical use case, experimental settings, and computational efficiency. While existing modules are used in the work, the overall quality of the work is high and solid. The authors should include the training budget in the paper, as it is important from a practical perspective, and further improve the presentation.